# Planetary-scale streak structure reproduced in high-resolution simulations of the Venus atmosphere with a low-stability layer

Hiroki Kashimura [1,2], Norihiko Sugimoto [3], Masahiro Takagi[4], Yoshihisa Matsuda[5], Wataru Ohfuchi [1,2], Takeshi Enomoto [6,7], Kensuke Nakajima[8], Masaki Ishiwatari[9], Takao M. Sato [10,11], George L. Hashimoto[12], Takehiko Satoh [10,13], Yoshiyuki O. Takahashi[1,2] & Yoshi-Yuki Hayashi [1,2]

Cloud patterns are important clues for revealing the atmospheric circulation of Venus. Recently, a planetary-scale streak structure has been discovered in middle- and lower-cloud images of Venus' night-side taken by IR2, the 2-μm camera, on board the Akatsuki orbiter. However, its formation mechanism has not been investigated. Here we succeed, for the first time, in reproducing the patterns of the observed streak structure, as regions of strong downward flows that develop in high-resolution global simulations of the Venus atmosphere. The streaks are formed in both hemispheres with equatorial symmetry, which is caused by equatorial Rossby-like and Kelvin-like waves with zonal wavenumber one. The low-stability layer that has been suggested by past observations is essential for reproducing the streak structure. The streaks of downward flow result from the interaction of the meridionally tilted phase lines of the Rossby-like waves and the characteristics of baroclinic instability produced around the low-stability layer.

[1] Center for Planetary Science, Kobe University, 7-1-48, Minatojima-Minamimachi, Chuo-ku, Kobe, Hyogo 650-0047, Japan. [2] Department of Planetology, Kobe University, 1-1, Rokkodai, Nada-ku, Kobe, Hyogo 657-8501, Japan. [3] Research and Education Center for Natural Sciences, Department of Physics, Keio University, 4-1-1, Hiyoshi, Kohoku-ku, Yokohama, Kanagawa 223-8251, Japan. [4] Department of Astrophysics and Atmospheric Science, Kyoto Sangyo University, Motoyama, Kamigamo, Kita-ku, Kyoto, Kyoto 603-8555, Japan. [5] Department of Astronomy and Earth Science, Tokyo Gakugei University, 4-1-1, Nukuikitamachi, Koganei, Tokyo 184-8501, Japan. [6] Disaster Prevention Research Institute, Kyoto University, Gokasho, Uji, Kyoto 611-0011, Japan. [7] Application Laboratory, Japan Agency for Marine-Earth Science and Technology, 3173-25, Showamachi, Kanazawa-ku, Yokohama, Kanagawa 236-0001, Japan. [8] Department of Earth and Planetary Sciences, Kyushu University, 744, Motooka, Nishi-ku, Fukuoka, Fukuoka 819-0395, Japan. [9] Department of Cosmosciences, Hokkaido University, Kita 10, Nishi 8, Kita-ku, Sapporo, Hokkaido 060-0810, Japan. [10] Institute of Space and Astronautical Science, Japan Aerospace Exploration Agency, 3-1-1, Yoshinodai, Chuo-ku, Sagamihara, Kanagawa 252-5210, Japan. [11] Space Information Center, Hokkaido Information University, 59-2, Nishinopporo, Ebetsu, Hokkaido 069-8585, Japan. [12] Department of Earth Sciences, Okayama University, 3-1-1, Tsushimanaka, Kita-ku, Okayama, Okayama 700-8530, Japan. [13] Department of Space and Astronautical Science, SOKENDAI, Shonan Village, Hayama, Kanagawa 240-0193, Japan. Correspondence and requests for materials should be addressed to H.K. (email: hiroki@gfd-dennou.org)

Venus is fully covered by thick clouds which hide most of its atmospheric circulation. However, the global patterns of clouds should reflect and be clues for revealing the atmospheric circulation. Observations of the Venus clouds by ground-based telescopes and spacecraft at various wavelengths have found some prominent features in the Venus atmosphere. At the cloud top level around 65 km height, a global Y-shaped structure was observed in the ultraviolet region[1] and a large bow-shaped structure was discovered in the mid-infrared region[2]. Spatial patterns of middle and lower clouds have been observed by utilising wavelengths near 1.74 μm and 2.3 μm, which correspond to spectral windows of the carbon dioxide atmosphere for the radiative emission from the hot (300–500 K) lower atmosphere at 20–30 km altitudes[3–12]. Contrast features in the night-side images at these wavelengths are believed to be due to spatially inhomogeneous opacity of dense middle and lower clouds with particle size of about 3.7 μm (called mode 3) located around 49–57 km heights[13,14]. The ground-based Anglo-Australian Telescope and the Galileo Near Infrared Mapping Spectrometer (NIMS) obtained Venus' night-side maps at these wavelengths and observed planetary-scale band-like structures, but their details were obscure because of low spatial resolutions[4–7]. The Visual and InfraRed Thermal Imaging Spectrometer (VIRTIS) on board Venus Express obtained high-resolution images at various infrared wavelengths, but observed locations were limited mainly to the southern polar region[8–11]. An image of a huge spiral pattern extending to the low-latitudes in the southern hemisphere was obtained[3], but its details have not been investigated. On December 7, 2015, the Venus Climate Orbiter/Akatsuki was inserted into the near-equatorial orbit circling around Venus[15–17]. High-resolution night-side global images including both hemispheres of Venus were obtained for the first time by IR2, the 2-μm camera[18,19], on board. The images taken by 1.74, 2.26 and 2.32 μm of IR2 revealed new details of Venus' night-side including narrow wavy ribbons, long dark streaks, sharp boundaries and even mesoscale vortices[12].

Numerical simulations can be useful for understanding the dynamics of observed features in the Venus atmosphere. One of the most successful Venusian atmospheric general circulation models (GCMs) is the AFES-Venus[20,21], which is a Venus version of AFES, the Atmospheric GCM for the Earth Simulator[22,23]. Unlike modern, sophisticated Venusian GCMs with realistic physical processes[24,25], the AFES-Venus is a simplified dynamical model similar to those used in many past Venusian GCM studies[26–28] which pursued generation of the superrotational state observed in the Venus atmosphere[29] from a motionless initial state with low spatial resolutions (~11- or 5.6-deg. intervals). The strategy of the AFES-Venus is to investigate possible circulation features of the Venus atmosphere with higher spatial resolutions (grid intervals less than about 2.8 deg.) compared to that of the previous studies but considering an idealised superrotation as an initial state. Despite simplifications of not including sophisticated radiative calculations, cloud processes, photochemical reactions, or topography, the AFES-Venus successfully reproduced superrotating zonal flow consistent with observations at the cloud level[21], and also the warm polar region surrounded by the cold latitudinal band or so-called cold collar[30]. The AFES-Venus has been used, more recently, to investigate three-dimensional structures of the polar vortex[31] and thermal tides[32]. A notable feature simulated in the AFES-Venus is a layer with low static stability, which was realized by a simplified radiative cooling (i.e., Newtonian cooling). Such a low-stability layer was suggested by the Pioneer Venus probe observations[29] and confirmed with recent radio occultation observations by Venus Express[33] and Akatsuki[34] to be located at the altitudes from about 50 to 60 km. The higher spatial resolution and the introduction of the low-stability layer would be the keys for the successful results of the AFES-Venus[35].

In this study, we introduce a night-side image of Venus at 2.26-μm wavelength obtained by IR2 to show the existence of a planetary-scale streak structure, and we perform four numerical simulations to reproduce and understand the streak structure by use of the AFES-Venus with a further high-resolution (about 0.75-deg. intervals). One is a standard run (STD) and the others are sensitivity experiments (ZS0, ZS2, and ZS4; see Methods for details of the experimental settings) to explore the dependence of the simulated streak structure on the presence of the diurnal cycle of insolation and that of the low-stability layer. A brief summary of our results and conclusions are as follows: The planetary-scale streak structure is reproduced as elongated regions of strong downward flow penetrating from above to near the bottom of the low stability layer (~50–60 km). The streak structure persistently develops with north-south symmetry for low- and mid-latitudes (≲75 deg.), even when simulated without the diurnal cycle of insolation. In the runs where the low-stability layer is eliminated, the streak structure disappears. The north-south symmetry is regulated by a zonal-wavenumber-one disturbance probably emerging from a vertical shear instability that consists of equatorial Rossby-like and Kelvin-like waves. The streak structure of the strong downward flows seems to be formed by the horizontal-wind convergence that is related to the meridional tilting of the polarmost sides of the equatorial Rossby-like wave. The tilting of the equatorial Rossby-like wave seems to be induced by the intense baroclinic disturbances that occur only in the cases where the low-stability layer exists. Therefore, we speculate that the streak structure might be a manifestation of front genesis caused by one or two types of baroclinic instability unique to the Venus atmosphere.

## Results

**Description of the streak structure**. A calibrated image of the night-side of Venus taken by IR2 is shown in Fig. 1. This image captures infrared radiation at 2.26 μm emitted by the hot atmosphere around 30 km altitudes and partly absorbed by dense middle and lower clouds lying at altitudes of around 49–57 km[13,14]. That is, bright colour in Fig. 1 indicates low-opacity regions and dark colour indicates high-opacity regions. One of the most prominent features in this image is the existence of a planetary-scale streak structure that is composed of many bright streaks located in both hemispheres with a rough equatorial symmetry. The streaks are enhanced and become clearer as

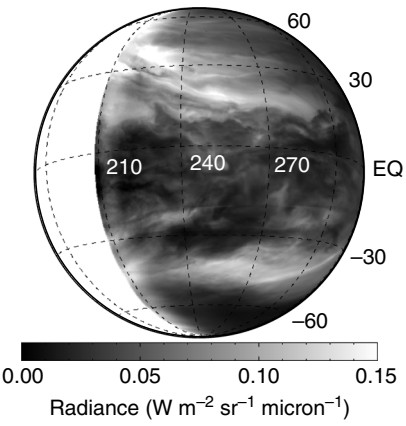

**Fig. 1** Calibrated image of the night-side of Venus taken on 25 March 2016 07:33 UT by IR2, the 2-μm camera, on board the Venus Climate Orbiter/ Akatsuki. Contrast features in the image are believed to reflect the opacity contrast of dense middle and lower clouds with a particle size of about 3.7 μm (called mode 3) located around 49–57 km height. Note that the sub-spacecraft longitude and latitude are 246.0 deg. and −9.1 deg., respectively

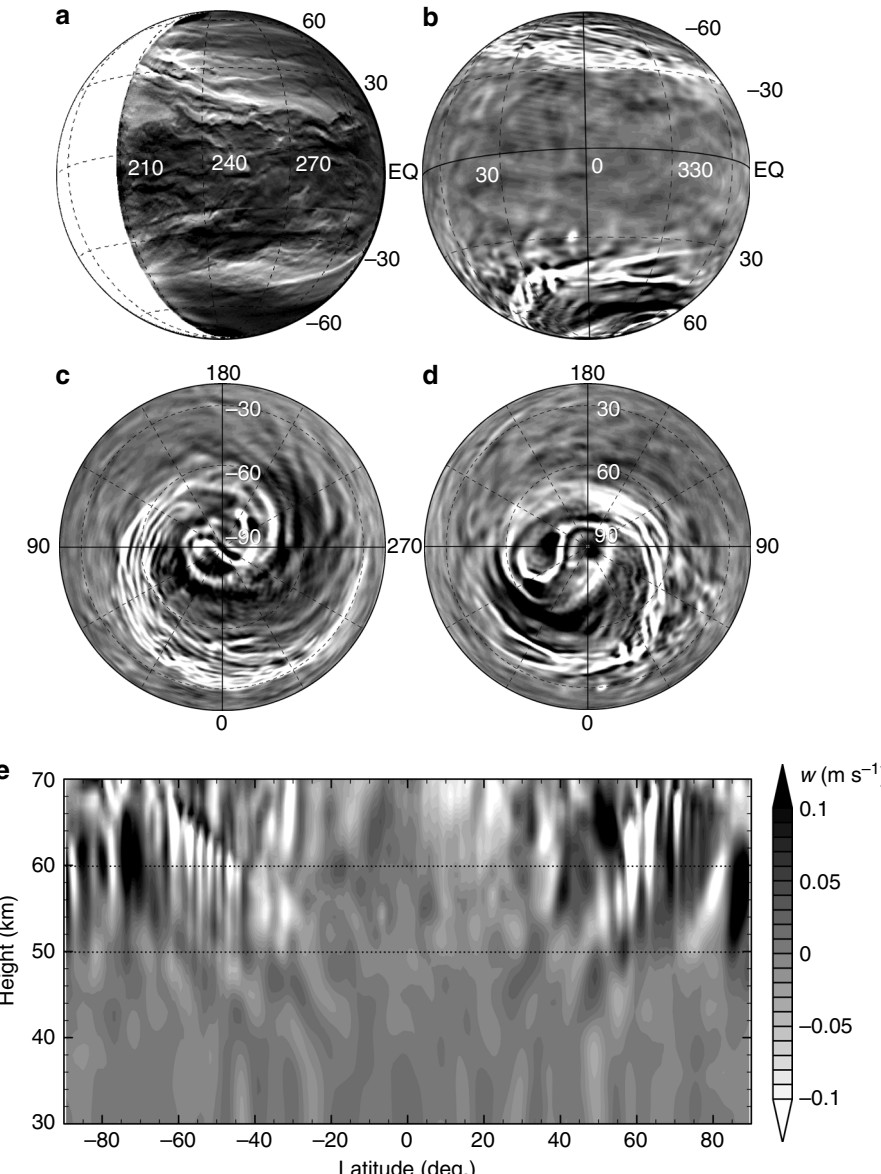

**Fig. 2** Morphology of the observed and the simulated planetary-scale streak structures. **a** Edge-enhanced image of Venus' night-side obtained from the image in Fig. 1; see Methods for the edge-emphasis process. **b** Vertical velocity at an altitude of 60 km simulated in STD after 3 Earth years (about 9.38 Venusian solar days) of time-integration, shown as a satellite view mimicking the panel **a**. **c**, **d** Same as **b** but for orthographic views from the south pole and the north pole, respectively, at the same altitude. **e** Same as **b** but for meridional cross-section at lon = 60 deg.; horizontal dotted lines indicate 50 and 60 km heights, which are the approximate boundaries of the simulated low-stability layer. Note that the planetary-rotation direction in the simulation is opposite to that of Venus, but the panel **b** is displayed with a 180-deg. rotation to easily be compared with the Venus image shown in the panel **a**. The colour bar is common for the panels from **b** to **e**

shown in Fig. 2a by an edge-emphasis process described in Methods. In the northern hemisphere, for instance, the streaks extend from the northwest to southeast, at least for 7700 km from (lon, lat) ~ (210, 35) to (290, 15), and the width of each streak is hundreds of km. Since downward flow is considered to decrease the cloud opacity, the streak structure in bright colour may indicate the presence of strong downward flow[19]. A further discussion on the relation between the lower-cloud opacity and downward flow is given in Discussion.

Figure 2b shows a simulated vertical velocity ($w$) field in the STD run at an altitude of 60 km mapped as a satellite view mimicking the IR2 image shown in Fig. 2a. Note that the planetary rotation direction in our simulation is opposite to that of Venus, so that Fig. 2b is rotated to be reasonably comparable

with Fig. 2a. Strong downward flows indicated by bright areas form a streak structure similar to that observed. The simulated structure is composed of several narrow streaks of downward flows and stretches from high-latitudes to low-latitudes of ±30 deg. in both hemispheres. In the lower hemisphere in Fig. 2b, the latitudinal locations of the streaks agree well with those in the southern part of the IR2 image; whereas, in the upper hemisphere, the simulated streaks are located slightly poleward compared with those in the northern part of Fig. 2a. There is a notable qualitative morphological similarity between the simulated and the observed streak structure, although there are quantitative differences between them. As shown later in Fig. 2e, the downward flows composing the narrow streaks at the altitudes of about 60 km penetrate down to the altitudes of

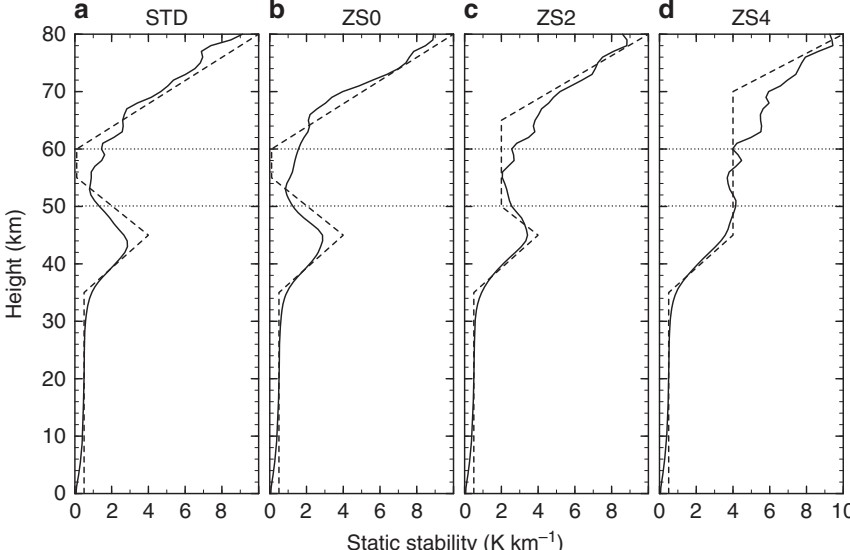

**Fig. 3** Vertical profiles of the static stability ($\Gamma - \Gamma_d$). The dashed line shows the static stability of the basic temperature profile for Newtonian cooling and solid line shows that of globally averaged simulated atmosphere in each run of **a** STD, **b** ZS0, **c** ZS2, and **d** ZS4. Horizontal dotted lines show 50 and 60 km heights

about 50 km. However, the streak structure at 50 km height is less clear as a whole. The discrepancy in altitudes between the IR2 image representing the features of around 49–57 km heights and our simulation will be discussed later.

Polar projections of simulated streaks are shown in Fig. 2c and d. In the polar regions (about >75 deg.) of each hemisphere, there is a dipole structure similar to the polar vortex observed by VIRTIS[10], as previously reported[30]. The streaks extend from the edge of the polar region and form huge spirals in each hemisphere; the morphology is similar to the spiral observed in the hemispheric image obtained by VIRTIS[3]. The simulated spirals and polar dipoles seem to be evolving independently. The simulated spirals in both hemispheres are persistently developed with mostly equatorial symmetry (i.e., they are synchronised with each other), while the simulated polar dipoles are not synchronised (see Supplementary Movies 1 and 2). The rotation period of the simulated dipoles is around 2–4 Earth days, while that of the simulated streak structure is about 6 Earth days. The rotation period of the mean zonal flow at this level is about 4.9 Earth days at the equator and is shorter in higher latitudes, so that the rotation period of the simulated streak structure is longer than that of the mean flow. The streak structures appear permanently in our simulation at least for 1 Earth year (about 3 Venusian solar days).

A snapshot of the meridional cross-section of vertical velocity (Fig. 2e) shows that downward flows of the streaks penetrate to the altitudes of around 50 km, where the mean static stability becomes as high as 1.2 K km$^{-1}$ as shown in Fig. 3a. Downward flows form slantwise from polar-upper regions to equatorward-lower regions. The polar view of the meridional velocity ($v$; northward is positive) field is shown in Fig. 4e, which is at the same level and time as in Fig. 2c. Regions of downward flow stronger than $-0.05$ m s$^{-1}$ are overlaid by green hatches. A sharp convergence zone of the meridional flow, shown as the boundary between equatorial-sided blue (poleward) regions and polar-sided red (equatorward) region, is formed and its location corresponds to that of the upper, equatorial side edge of the downward flow streaks. The meridional cross-section (Fig. 4i) shows that the sharp convergence zone is also tilted poleward as height increases and corresponds to the starting points of the downward flows.

Consequently, at a given height level, several narrow streaks appear side by side as shown in Fig. 2b.

**Importance of the low-stability layer**. Sensitivity experiments are performed to explore the importance of the diurnal heating and the low-stability layer simulated in STD. The conducted simulation runs are listed in Table 1 and their details are described in Methods. The planetary-scale streak structure also appears in the ZS0 run that includes the same radiative cooling profile as in STD to represent the low-stability layer but excludes the diurnal variation of the solar heating as shown in Fig. 4b. In addition, the streaks in both hemispheres in ZS0 also rotate synchronously (see Supplementary Movie 3), and polar disturbances rotate faster and independently as in STD. The meridional velocity field in ZS0 is also similar to that in STD as shown in Fig. 4f and j. These results indicate that the diurnal variation of the solar heating does not play an essential role in the formation of the streak structure or its inter-hemispheric synchronisation. This might be surprising because the solar diurnal variation seems to be the largest, equatorially symmetric external forcing (at least in the altitudes of 50–70 km) in the Venus atmosphere. The inter-hemispheric synchronisation of the planetary-scale streak structure in ZS0 suggests that the streak structure exists in the Venus atmosphere irrespective of the local time through the Venusian solar day.

In contrast to the solar diurnal heating, the low-stability layer is crucial for the appearance of the streak structure. The static stability to form the low-stability layer given in the basic temperature profile for the Newtonian cooling is increased from 0.1 K km$^{-1}$ of STD/ZS0 to 2.0 K km$^{-1}$ in ZS2 and to 4.0 K km$^{-1}$ in ZS4, respectively as shown by dashed lines in Fig. 3. Note that these given static-stability profiles are used for the basic temperature for the Newtonian cooling, whereas the simulated ones shown by solid lines in Fig. 3 are results of thermal balance among the Newtonian cooling, solar heating, and adiabatic heating/cooling due to atmospheric circulations. In ZS2 and ZS4 the simulated static stability do not reach as low as 1.0 K km$^{-1}$ around 50–60 km heights. In ZS2, the area of strong downward flow still exists but is no longer concentrated or elongated and

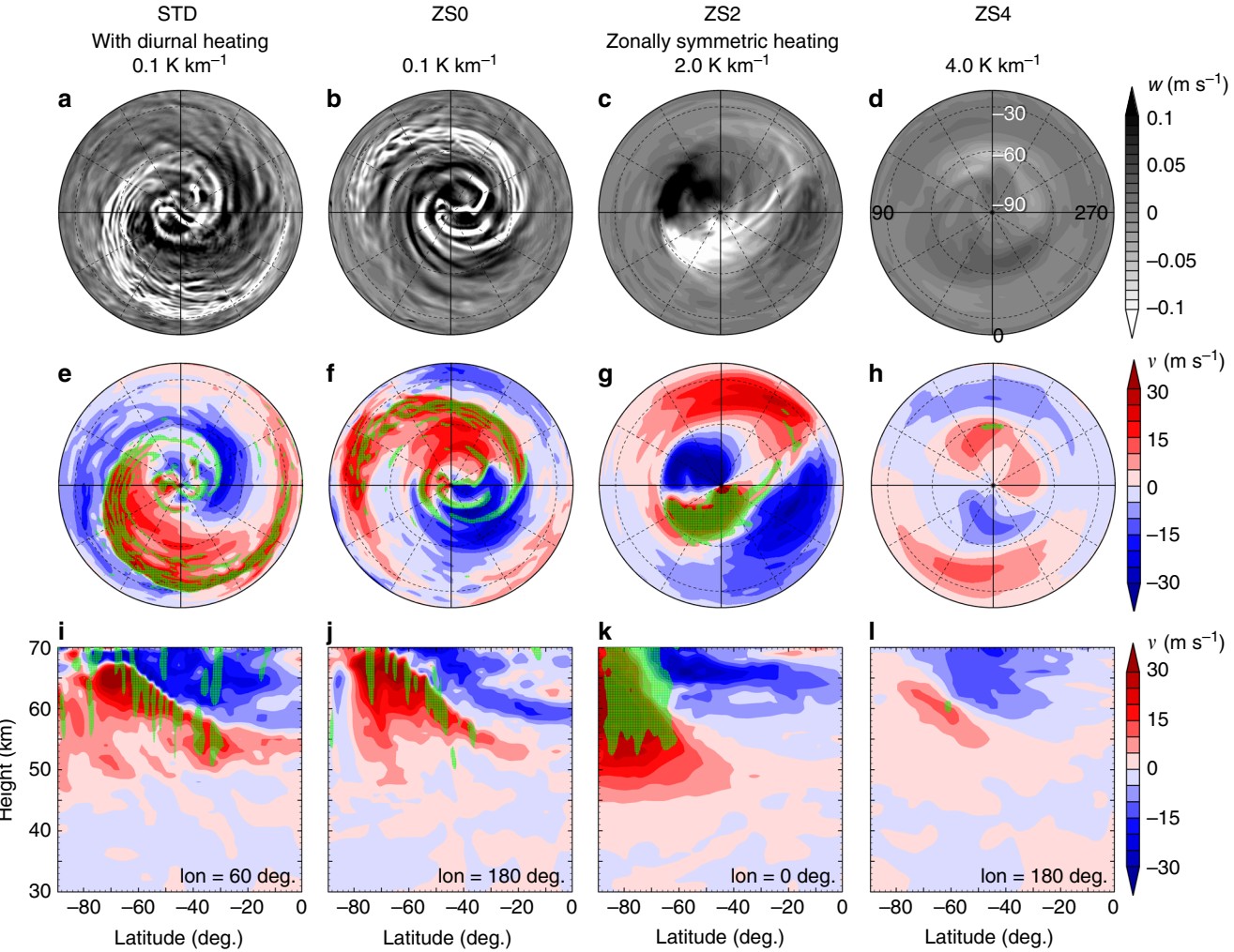

**Fig. 4** Results of the numerical experiments. Settings are shown on the top of the figure. Panels **a–d** show snapshots of vertical velocity at the altitude of 60 km, and **e–h** show those of meridional velocity seen from the south pole in the model; **i–l** are meridional cross-sections of meridional velocity. Green hatching indicates strong downward flow ($w < -0.05$ m s$^{-1}$). In **i–l**, the longitude of the meridional cross-section is shown in each panel

**Table 1 Experimental settings of the simulation runs**

| Run | Solar heating | Stability (K km$^{-1}$) | Altitude range (km) |
|-----|---------------|-------------------------|---------------------|
| STD | with diurnal variation | 0.1 | 55–60 |
| ZS0 | zonally symmetric | 0.1 | 55–60 |
| ZS2 | zonally symmetric | 2.0 | 50–65 |
| ZS4 | zonally symmetric | 4.0 | 45–70 |

The right two columns show the static stability and the altitude range of the low-stability layer given in the basic temperature profile for Newtonian cooling shown by dashed lines in Fig. 3. See Methods for the details of the solar heating and the low-stability layer

cannot be recognised as a streak structure (Fig. 4c); the convergence region of meridional velocity is confined in high-latitudes and does not extend to mid- or low-latitudes (Fig. 4g and k). In ZS4, the highest-stability run, vertical velocity is much weaker than in other runs (Fig. 4d), meridional velocity and its convergence are also weak (Fig. 4h and l), and no feature like the streak structure can be found.

The true nature of the streak structure (i.e., it's formation mechanism) remains to be identified. The upper panels in Fig. 5 show mean heat transport by eddies $\left(\overline{\rho v'\theta'}, \overline{\rho w'\theta'}\right)$ in each run, where $\rho$ is the density, $\theta$ is the potential temperature, prime

indicates deviation from the zonal mean, and overline indicates zonal- and time-mean. There is a large amount of poleward and upward heat transport, indicating the presence of baroclinic instability, in the STD and ZS0 runs shown by vectors in Fig. 5a and b, respectively. These heat fluxes can be classified into two groups by their locations. One is in the mid-latitudes (from $-30$ to $-60$ deg.) at the altitudes of about 55–70 km; and the other in the high-latitudes (from $-60$ to $-80$ deg.) at the altitudes of about 50–65 km. Such two-grouped eddy heat fluxes are also shown in Fig. 3b of Sugimoto et al.[21] with diurnal solar heating, though they did not explore them separately. With the increase of the static stability, the vertical component of the eddy heat transport weakens in ZS2 (Fig. 5c), and the entire heat transport by eddies almost disappears in ZS4 (Fig. 5d).

Increasing the static stability of the low-stability layer also brings a significant change in the mean (Ertel's) potential vorticity (PV) field as shown in Fig. 5e–h. In the southern hemispheres in STD and ZS0, the mean PV fields are rather complicated but have common characteristics; slightly negative or even positive PV are widely distributed in the mid-latitudes of about 50–60 km heights and polar region of about 55–60 km. Here, not only the lowness of the stability but also the horizontal component of vorticity contributes to produce positive values of PV in the southern hemisphere. In contrast to the low-stability cases, such local maxima of PV do not exist in the high-stability

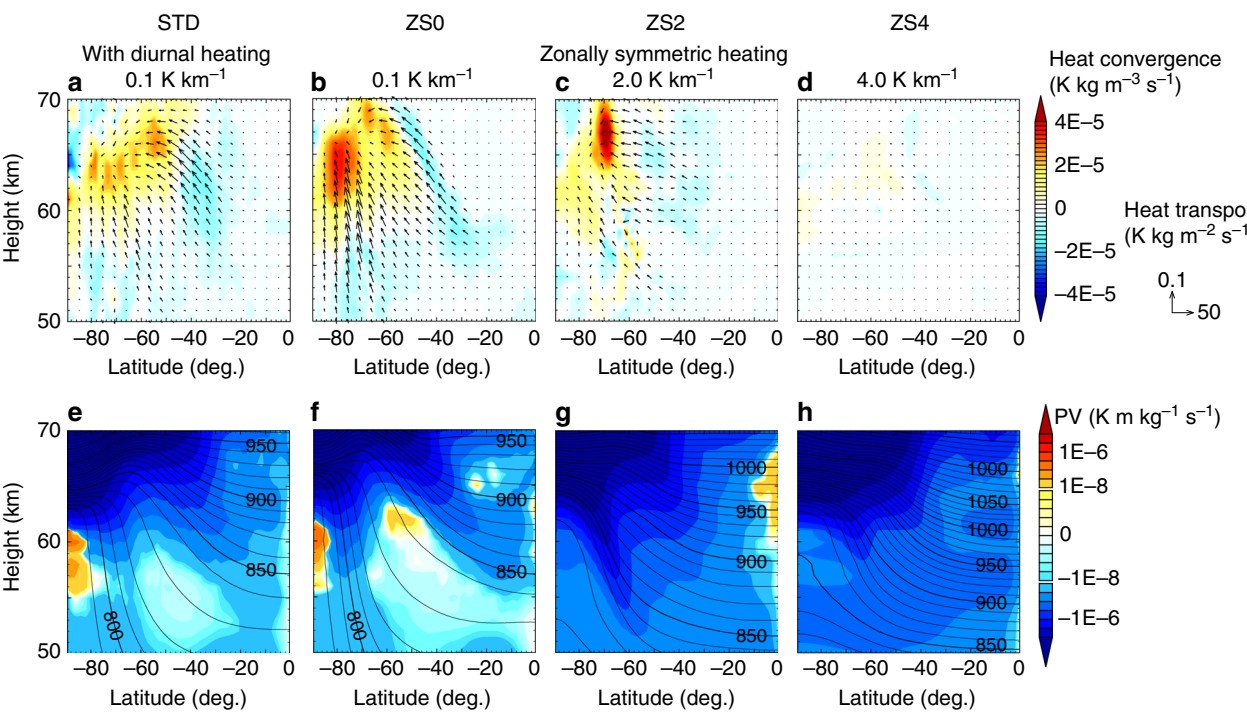

**Fig. 5** Zonally and temporally averaged fields of eddy heat transport and potential vorticity (PV). Panels **a**–**d** show heat transport due to eddies by vectors and its convergence by colour, and **e**–**h** show PV by colour and potential temperature by contour. For the colouring of PV, a linear scale is used for the range between ±10$^{-8}$ and a logarithmic scale is used for outside of the range to visualise the positive and negative PV-gradients. The time-average period is 30 Earth days (about 5–7 periods of the superrotation at 60 km height)

cases shown in Fig. 5g and h; note that positive PV in the equatorial region would be due to fluctuation across the equator and may disappear by taking much longer time-averaging period. As for the emergence of the baroclinic instability, a mean PV-gradient along isentropes must reverse in the vertical direction[36]. The isentropic PV-gradient is mostly positive (PV increases with latitude). However, in the mid-latitudes of STD and ZS0, there are regions of negative isentropic PV-gradient on the equator- and lower-side of the PV maxima, which indicates a possibility of baroclinic instability caused by the coupling of disturbances on different levels of isentropes with opposite sign of PV-gradient. Similarly, in the polar regions of STD and ZS0, there appears negative PV-gradient along the isentropes close to the polar PV maxima. In contrast to the low-stability cases, there appear almost no regions of negative isentropic PV-gradient in ZS2 and ZS4; especially in the mid-latitudes, PV increases quite monotonically with latitude. This indicates that the necessary condition for baroclinic instability is not satisfied in those higher-stability cases. These PV trends confirm that the existence of the low-stability layer may cause baroclinic instability as previously reported in simulation studies of Venus atmosphere[20,21,37], and suggest that intense baroclinic disturbances are essential for the formation of the streak structure. However, the resulting disturbance structures seem to be more complicated than formerly expected and are left to be investigated in our future work (see Discussion).

**Horizontal structure regulating the north-south symmetry.** A longitude-time cross-section (as known as Hovmöller diagram) of pressure anomaly from the zonal mean ($p'$) at 35 deg. south latitude and 65 km height in ZS0 is shown in Fig. 6a. Here,

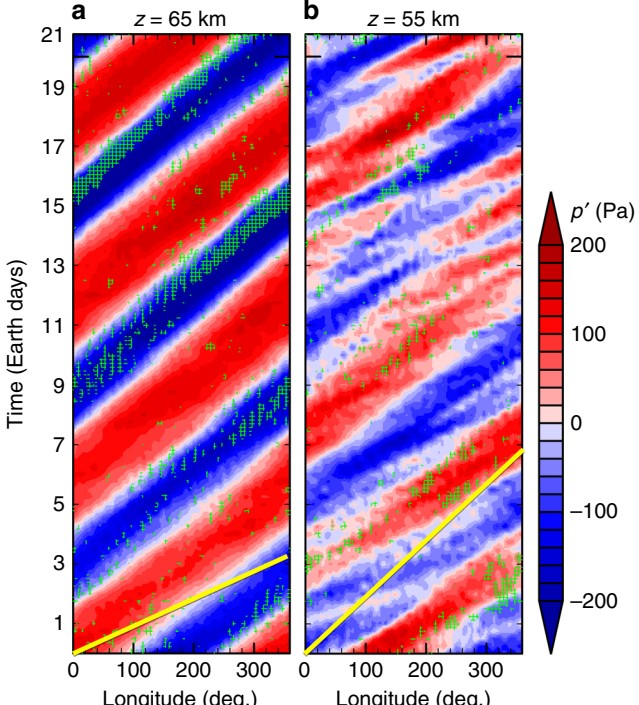

**Fig. 6** Longitude-time cross-sections (Hovmöller diagram) of the pressure anomaly (red-blue shading) at the latitude of −35 deg. for ZS0. The altitudes are **a** 65 km and **b** 55 km. Green hatching shows regions of downward flow stronger than −0.03 m s$^{-1}$. The yellow solid line shows the zonal mean zonal velocity at each height

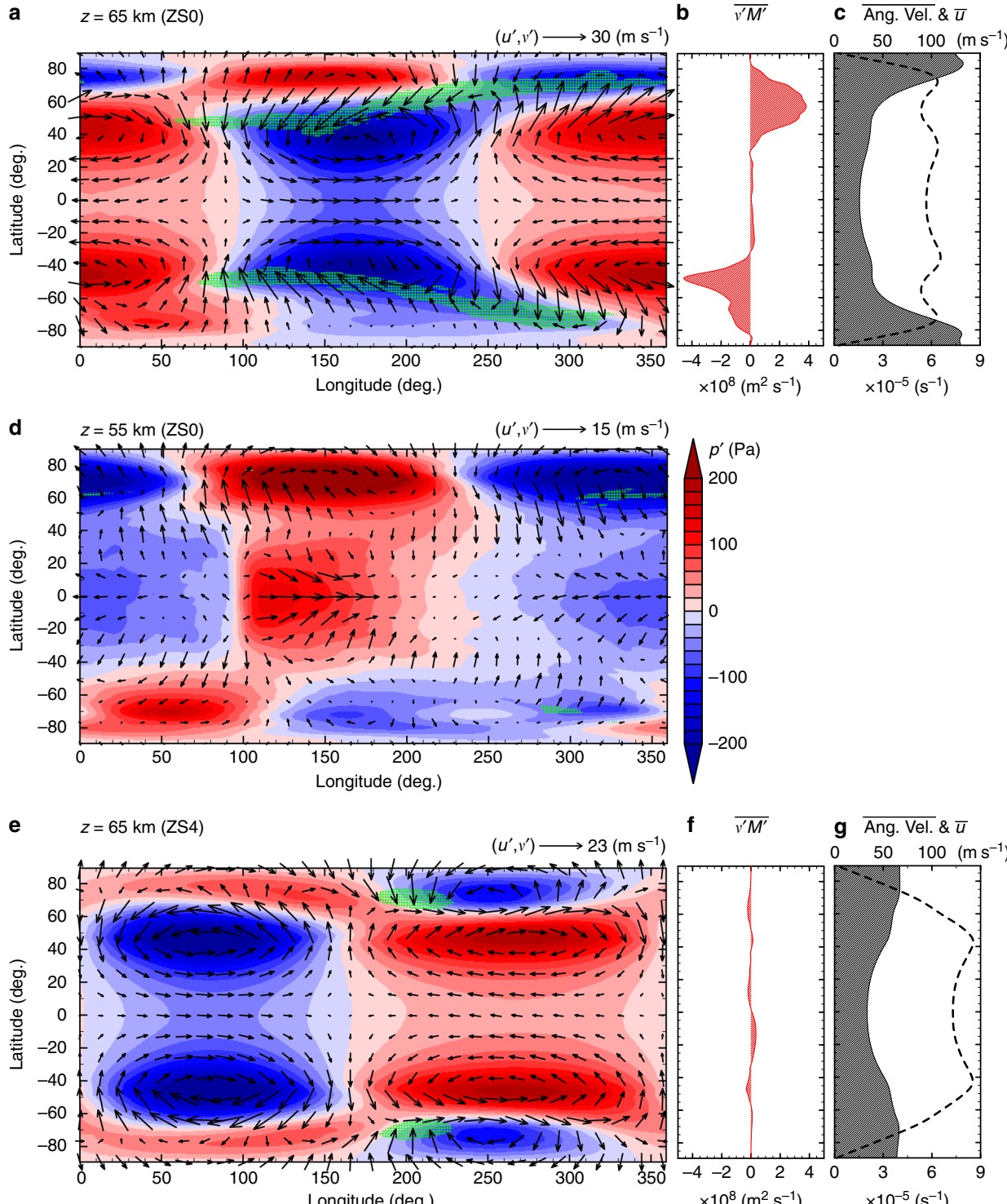

**Fig. 7** Composite means of pressure anomaly $p'$ (red-blue shading) superimposed by horizontal flow anomaly $(u', v')$. Panels **a**, **d** show the composite means on $z = 65$ km and 55 km, respectively, for ZS0; **e** shows that on $z = 65$ km for ZS4. Composite means are taken along the propagation speeds associated with the dominant Rossby-like waves at $z = 65$ km, the equatorial values of which are 75.6 m s$^{-1}$ for ZS0 and 96.0 m s$^{-1}$ for ZS4, respectively. Green hatching indicates strong downward flow ($w < -0.03$ m s$^{-1}$). **b**, **f** Temporal and zonal mean meridional transports of angular momentum by eddies at 65 km height for ZS0 and ZS4, respectively. The period of time-average is the same as that in Fig. 5. **c**, **g** Same as **b** and **f** but for angular velocity (shaded solid line, scales written in bottom axis) and zonal wind (thick dashed line, top axis)

assuming that the streak structures in STD and ZS0 are caused by the same mechanism, we analyse ZS0, which does not have the influence of the diurnal heating, to investigate the reason for the north-south symmetry of the structure. The latitude and height used for Fig. 6a are where the amplitude of an equatorial

Rossby-like wave explained below is large in ZS0. The dominant pressure anomaly has a longitudinal structure of wavenumber one and coherently moves eastward with a rotation period of about 5.8 Earth days (about 62.1 m s$^{-1}$ at this latitude; remember that the direction of the planetary rotation is eastward in our

simulation). It is slower than the mean flow rotating with a period of about 3.3 Earth days (about 109.3 m s$^{-1}$) at this latitude shown by the yellow solid line. The dominant pressure anomaly at this level propagates westward relative to the mean flow. The strong downward flow shown by green hatches also propagates with the same speed as the pressure anomaly. In Fig. 6b, $p'$ at the lower altitude (55 km height) is plotted. We can recognise a clear signal in $p'$, though it is noisy compared to the upper level, which propagates with nearly the same speed as that at 65 km. Now, the rotation period of the mean flow at 55 km height is about 6.8 Earth days (about 53.1 m s$^{-1}$), which is slower than the disturbance. That is, the dominant pressure anomaly at 55 km propagates eastward relative to the mean flow.

In order to investigate the disturbance structure, we took composite means of pressure disturbance $p'$, horizontal wind disturbance $(u', v')$ and vertical flow $w$ following the propagation of the dominant pressure signal at 65 km height shown in Fig. 6a. The composite means highlight the coherent structure associated with the dominant signal by averaging the snapshots of the fields shifted zonally according to its propagation. The composited structure of $p'$ and $(u', v')$ on 65 km height in the low- and mid-latitudes (<60 deg.) shown in Fig. 7a resembles that of the equatorial Rossby wave[38] with wavenumber one; i.e., two pairs of low- and high-pressure anomalies associated with cyclonic and anti-cyclonic circulations, respectively, are straddling the equator. On the other hand, in the lower altitude at 55 km height, the composited structure in latitude <50 deg. resembles that of the equatorial Kelvin wave[38] with wavenumber one (Fig. 7d); i.e., a low- or high-pressure anomaly is located on the equator associated with intense westward or eastward flow anomaly, respectively, while meridional flow is weaker than zonal flow. Note that one of the reasons why the Hovmöller diagram of Fig. 6b is noisy is that the plot is not at the equator but at 35 deg. south latitude where there are influences of other disturbances.

Recall that the propagation speed of $p'$ is nearly the same at both levels as shown in Fig. 6. This implies an occurrence of vertical shear instability by the combination of the equatorial Rossby-like wave around $z = 65$ km and the equatorial Kelvin-like wave around $z = 55$ km[39,40]. Coincidence in the zonal distribution of $u'$ in the equatorial region on both levels (eastward in 100–240 deg. and westward in the other longitudes) shows that these waves are coupled with and maintained by each other. The vertical shear instability that contains the structure of the equatorial Kelvin-like wave would result in the north-south symmetry or synchronisation of disturbances in the low- and mid-latitudes in the hemispheres.

Temporal and spatial structures similar to those shown in Fig. 6 and 7 are also obtained in STD where the diurnal solar heating is included (not shown). Though the thermal tide of wavenumber one appears with a noticeable amplitude in the snapshots of the pressure anomaly field, it disappears after taking the composite mean and has no significant effects on the horizontal structure described above.

In Fig. 7, there are strong signals in the polar regions (>75 deg.). However, the north-south symmetry no longer holds there. The polar anomalies in both hemispheres develop independently from the symmetric equatorial waves at least in STD and ZS0. In other words, the polar pressure anomalies could prevent the equatorial waves and the streak structure from reaching the polar regions. We should note that, as mentioned earlier, those polar disturbances propagate faster, and they should eventually vanish in the composite fields if we took the average over a far longer time period. We should also note that the structure of polar anomalies is quite deep ranging over the altitudes from 40 to 70 km roughly along an isentropic surface (shown in Supplementary Fig. 1). Since these polar disturbances seem to have

locally uniform structure in the vertical direction, the instability generating them might be interpreted as barotropic instability as previously reported[31]. However, because of the presence of the meridional eddy heat fluxes shown in Fig. 5, we suggest that the high-latitude baroclinic instability that is separated from the mid-latitude baroclinic instability also contributes to the polar disturbances (see also Discussion and Fig. 8).

**Key features for the formation of the steak structure**. A characteristic feature corresponding to the streak structure is a meridional tilting of the phase of the equatorial Rossby-like wave (Fig. 7). In ZS0, the polar-side halves (in latitudes from 45 to 65 deg.) of the equatorial Rossby-wave-like structure in the pressure anomalies ($p'$) and eddy circulations ($u', v'$) are tilted eastward with increasing latitude as shown in Fig. 7a. The tilted $p'$ and $(u', v')$ seem to extend poleward and intrude into the polar anomalies appearing in latitudes around 75 deg. Such tilting and extension of the circulations seem to be related with the horizontal-wind convergence that forms the streak structure of the strong downward flow in the low-pressure regions as shown by green hatches. Conversely, in the high-stability case (ZS4), although an equatorial Rossby-like wave with zonal wavenumber one still emerges as seen in Fig. 7e obtained by taking a similar composite average as that of ZS0, the pressure anomalies are not meridionally tilted and the downward flows are weak. The difference in tilting of the eddy circulations between ZS0 and ZS4 is also apparent in angular-momentum transport by eddies shown in Fig. 7b and f. A large amount of poleward transport of angular momentum exists in ZS0, whereas the angular momentum transport is very small in ZS4. Correspondingly, another characteristic feature observed in ZS0 contrasted with ZS4 is the existence of high-latitudinal zonal jets (local maximum in the zonal mean zonal wind) around 75 deg. as shown by dashed lines in Fig. 7c. In terms of angular velocity, a large meridional shear emerges in high-latitudes in ZS0 (Fig. 7c), but not in ZS4 (Fig. 7g). Recall that the intensity of baroclinic disturbances is strong in ZS0 and quite small in ZS4 as shown in Fig. 5b and d, the baroclinic instability must play an important role for producing those features. The structure of the baroclinic disturbances in the model, however, is yet to be fully investigated. We just try to discuss the possible structure briefly in the next section.

## Discussion

Because baroclinic disturbances are associated with strong meridional heat fluxes in the vicinity of zonal jets, we turn to an analysis of these features in our simulations. Fig. 8 shows the mean meridional distribution of the north-south eddy heat flux $(\overline{\rho v' \theta'})$ and the mean zonal wind $(\bar{u})$ in ZS0. There are two axes of large poleward heat fluxes, each of which implies the emergence of baroclinic instability: one in the mid-latitudes indicated by the white dashed line and the other in the high-latitudes indicated by the black one, in each hemisphere. The locations of the high-latitude heat fluxes align with those of high-latitude jets. This implies that baroclinic eddies in high-latitudes contribute to the formation of the jet by PV stirring[41,42], the same manner as that of the mid-latitude jets in the Earth's troposphere. Then, the high-latitude jets may incline the phase line of the equatorial Rossby-like wave. In contrast to the high-latitudes, contribution of the mid-latitude baroclinic disturbances is unobvious. The axes of jet maxima do not correspond to the maxima of the mid-latitude heat fluxes. Instead, a large amount of eddy angular-momentum fluxes exists over this region as shown in Fig. 7b and indicated by red arrows in Fig. 8 at the altitude of 65 km. We suspect that the lower-level part of the mid-latitude baroclinic disturbances (i.e., Rossby wave) appear as the signals propagating,

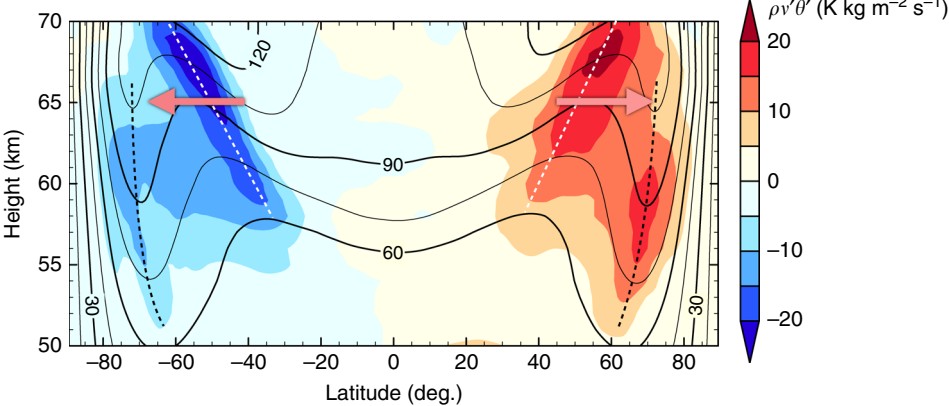

**Fig. 8** Zonally and temporally averaged distributions of zonal winds (contour) and north-south eddy heat fluxes (colour) in ZS0. White dashed lines indicate locations of large poleward heat fluxes suggesting the emergence of baroclinic instability in mid-latitudes; black dashed lines indicate those in high-latitudes and the axes of high-latitude jets. Red arrows show approximate latitudes of large poleward angular momentum fluxes by eddies at the altitude of 65 km, which is shown in Fig. 7b. The time averaging period is the same as Fig. 5 and contour interval is 15 m s$^{-1}$

with a period of about 3 Earth days, faster than the equatorial Kelvin-like wave's dominant disturbances whose rotation period is about 5.8 Earth days in Fig. 6b. The signals also propagate faster than the mean flow. This is consistent with the direction of the Rossby wave's phase speed under the negative PV gradient shown by Fig. 5f. Then, the following three components might be related to the formation of the streak structure: the high-latitude disturbances and the associated jets, the mid-latitude baroclinic disturbances, and the equatorial Rossby-like waves. The latitudinal angular phase speed of the high-latitude disturbances is the fastest, then that of the mid-latitude disturbance is in the middle, and that of the equatorial disturbance is the slowest. The differential phase speeds of those three components might cause the meridional tilting of the phase lines of the equatorial Rossby-like wave to form the streak structure; and the north-south symmetry is regulated by the vertical shear instability with the equatorial Rossby-like wave and the equatorial Kelvin-like wave.

Though we have discussed a possible formation mechanism of the planetary-scale streak structure as above, we should note that the details of the disturbances, instabilities, and the angular momentum balance in our simulation are still unclear and remain to be explored. The situations under which those disturbances took place in the simulated Venus atmosphere, e.g., absence of surface, vertical profile of the static stability, and importance of relative vorticity playing a role of planetary vorticity in a slowly rotating planet, differ largely from those in the Earth's troposphere. The structure of the waves and instabilities in our simulation should be quite different from those commonly considered in the Earth's meteorology. Interactions or relations among the baroclinic instability in the high-latitudes, that in the mid-latitudes, and the shear instability in the equatorial region are also open questions. Furthermore, the mechanism for the disappearance of the intense baroclinic disturbances in the higher-stability cases should be investigated. We need to understand these mechanisms to evaluate the robustness or sensitivity of the speculated formation mechanism presented here. However, we keep these further investigations for our future studies.

In the rest of this section, we discuss the discrepancies in altitudes between the observed and the simulated streak structures with remarks on the difference of physical quantities and simplifications adopted in our simulation. Although there is a remarkable similarity between the planetary-scale streak structure observed in the IR2 night-side image and that represented as strong downward flow in our simulation, the altitudes of the structure differ between the observed and the simulated. The

contrast features in the IR2 night-side image with 2.26 μm are considered to reflect the opacity contrast in the middle and lower clouds located around 49–57 km altitudes[13,14], the bottom and inside of the low-stability layer. On the other hand, the numerical model simulates the streak structure in the downward flow and the structure is clear at just above the low-stability layer ($z =$ 60 km in STD and 65 km in ZS0).

As shown in Fig. 2e the downwelling flow composing the streak structure penetrates into the low-stability layer. In addition, a penetration depth of an air parcel ($H$) can be roughly estimated as $H = W \times T \sim 15$ km by using $W = 0.1$ m s$^{-1}$ for the typical scale of $w$ and $T = L/U = 7.5 \times 10^6$ m/50 m s$^{-1} = 1.5 \times 10^5$ s for the timescale estimated from scales of the horizontal length ($L$) and propagation speed ($U$) of the streak structure. Hence, the vertical displacements due to the streak structure formed above the low-stability layer would reach deep into the low-stability and middle- and lower-cloud layers.

The dynamical downward displacement in the cloud layer may decrease the middle- and lower-cloud opacity. The downward flow tends to prevent the supply of sulphuric acid vapour, which might be essential for the generation of the middle and lower clouds[43], from the sub-cloud reservoir. Moreover, the air parcels displaced from the upper clouds include smaller cloud-particles generated at the cloud-top altitude (~65 km)[44] and push out old air which contains the larger, so-called mode 3 cloud-particles (about 3.7 μm). These contribute to the decrease of the opacity at 2.26-μm wavelength.

We should emphasise that our simulations are completely dynamical ones and do not contain sophisticated radiative calculation, photochemical reactions, or cloud processes, which should influence the cloud opacity observed in the night-side image taken by IR2. It is not easy to estimate the effects of these simplifications even qualitatively because there are many unknown aspects in the above processes in the Venus atmosphere. The cloud opacity, for example, should give a feedback to the flows because the radiative solar heating should depend also on the cloud opacity; while the solar heating is simplified to the function of the height and solar zenith angle in our model. The radiative calculation is also important because it may affect the stability and/or location of the low-stability layer, which is important for the formation of the streak structure; while the low-stability layer was mimicked through Newtonian cooling toward the externally given profile in our simulation. Despite such simplifications, the planetary-scale streak structure is reproduced even in the present dynamical setup. This fact suggests that

large-scale processes, rather than radiation or microphysics, are dominant for the formation mechanism of the streak structure in the Venus atmosphere.

The model's idealised settings of geometry may also affect some features of the streak structure. For example, the equatorial symmetry appears stronger in our simulation than in the IR2 image (Fig. 2a). Although the integration time might not be sufficient to check the presence of internal variability, this could be due to the lack of topography or the slight tilt of the rotation axis. The persistence of the streak structure is also strong in our simulation, while it has not been confirmed by observations. The streak structure in Venus could be less persistent because intermittent disturbances such as topographic gravity waves might disturb or conceal the structure. Effects of realistic geometry and topography on the streak structure need to be explored.

Our results should be considered as a qualitative dynamical interpretation of the streak structure. The quantitative aspects in this study such as vertical location and width of the streaks do not necessarily need to agree with observational facts. In order to confirm the interpretation of the streak structure observationally, explorations of a large amount of various data obtained by the Akatsuki observation would be useful. Data-assimilation of the Venus atmosphere[45] using Akatsuki's data would largely contribute to both qualitative and quantitative understanding of the streak structure, which is now being carried out along with new implementations of physical processes.

## Methods

**Edge-emphasis process.** The edge-emphasis process applied to Fig. 1 to produce Fig. 2a is as follows: First, Gaussian smoothing with a standard deviation of seven pixels is applied in the y-axis direction to the calibrated image shown in Fig. 1. Then, the smoothed image multiplied by 0.78 (the factor is experimentally determined) is subtracted from the calibrated image, and the edge-emphasised image is obtained. Note that, the day-side is white-shaded in both Figs. 1 and 2a.

**Numerical model and experimental settings.** For the Venus atmospheric simulation in this study, the primitive equations of a dry ideal gas are numerically solved by using a general circulation model, named AFES-Venus[20,21], which is a Venus version of AFES (Atmospheric GCM for the Earth Simulator)[22,23]. A spectral method with the spherical harmonics is used for the horizontal discretisation with the triangular truncation number of 159; the number of grid points for calculating nonlinear terms is 480 (longitude) × 240 (latitude) with about 0.75-deg. intervals. The vertical domain extends from the ground to the altitude of about 120 km discretised into 120 layers with an interval of about 1 km using $\sigma$-coordinate. Note that the AFES-Venus is a dynamical model of the Venus atmosphere and does not contain cloud processes, photochemical reactions, or topography. The radiative process is simplified as described later.

The following physical parameter values of Venus are used; the planetary radius is 6050 km, the gravity acceleration is 8.87 m s$^{-2}$, the rotation period is 243 Earth days, and the standard surface pressure is $9.2 \times 10^4$ hPa. The specific heat at constant pressure and the vertical eddy diffusion coefficient are assumed to be constant: 1000 J kg$^{-1}$ K$^{-1}$ and 0.15 m$^2$ s$^{-1}$, respectively. The biharmonic horizontal diffusion is included for dissipation. The damping time for the maximum wavenumber is 0.01 Earth days for assuring the stability of computation. A sponge layer which reduces eddy components of the flows and temperature is employed above 80 km to prevent wave reflection. The surface friction is represented by Rayleigh friction with damping time of 0.5 Earth days at the lowest layer. In our simulation, the direction of the planetary rotation is the same as that of the Earth (from west to east) and is opposite to that of Venus.

The radiative forcing in the infrared region is simplified by Newtonian cooling that relaxes the temperature to a horizontally uniform reference-temperature profile; its static stability is based on Tomasko et al[46]. For the standard run, as shown by the dotted line in Fig. 3a, the static stability $\Gamma - \Gamma_d$ is 0.5 K km$^{-1}$ below 35 km, increases linearly with height to reach the local maximum of 4 K km$^{-1}$ at 45 km, then decreases linearly to reach 0.1 K km$^{-1}$ at 55 km. Here, $\Gamma$ is the temperature lapse rate and $\Gamma_d = -g/C_p = -8.87$ K km$^{-1}$ is the dry adiabatic lapse rate. The static stability from 55 to 60 km is kept as low as 0.1 K km$^{-1}$ to mimic the low-stability layer. Above 60 km, the stability linearly increases with height to reach 10 K km$^{-1}$ at 80 km, then keeps constant up to the model top. The vertical profile of the coefficient of Newtonian cooling is adopted from Crisp[47]; the relaxation time decreases almost exponentially from about 10,000 Earth days at the surface to 0.1 Earth days at 120 km height. Solar heating is included on a basis of the vertical profile of solar flux determined from observations by Tomasko et al[46] and the horizontal structure proposed by Crisp[47]. The effect of solar flux absorbed by the ground surface is ignored for simplicity. For the standard run (STD), the horizontal structure of the solar heating on the day side is assumed to be proportional to $\cos^{0.6}\theta_s$ above 71 km height and to $\cos^{1.5}\theta_s$ in the lower layers, where $\theta_s$ is solar zenith angle, as proposed by Crisp[47].

An idealised superrotation whose zonal wind speed reaches 100 m s$^{-1}$ at 70 km height at the equator, very simply mimicking observed zonal-wind profiles[48], is introduced as the initial state. The zonal wind increases linearly from the ground and in solid-body rotation at each layer. The initial temperature field has a meridional gradient satisfying the thermal wind balance with the zonal flow. The static stability of the initial temperature is the same as that of the reference temperature of Newtonian cooling. Time-integration is performed by using the leap-frog method with the Asselin[49] filter. After the time-integration of about two Earth years (about 6.25 Venusian solar days), the simulated atmosphere mostly achieved a statistically steady state, and the data obtained in the fourth Earth year (after about 9.38 Venusian solar days) is used in this study.

In addition to the standard run (STD), three runs named ZS0, ZS2, and ZS4 were conducted for sensitivity experiments (Table 1). They do not include the diurnal variation of the solar heating and the zonally averaged solar heating profile is used. As shown by dashed lines in Fig. 3, the reference static-stability profile for Newtonian cooling in ZS0 is the same as that of STD, whereas in ZS2 (ZS4) the static stability of 2.0 (4.0) K km$^{-1}$ is used for the altitude range of 50–65 (45–70) km to eliminate the low-stability layer. The data-output interval was 6 h in ZS0, ZS2, and ZS4, while it was 24 h in STD.

**Code availability.** The model code of the AFES-Venus used in this study is available from the corresponding author on reasonable request.

## Data availability
The datasets generated and/or analysed during the current study are available from the corresponding author on reasonable request.

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

## Acknowledgements

We thank all of the members and contributors of the Akatsuki project. This study was partly conducted under the Earth Simulator Proposed Research Projects titled Simulations of Atmospheric General Circulations of Earth-like Planets by AFES and High Resolution General Circulation Simulation of Venus and Mars Atmosphere using AFES; and the simulations were performed on the Earth Simulator with the support of JAMSTEC. This study was also supported by MEXT as Exploratory Challenge on Post-K computer (Elucidation of the Birth of Exoplanets [Second Earth] and the Environmental Variations of Planets in the Solar System). This study was supported by JSPS KAKENHI 16H02225, 16K17809, and 17H02961. Data analyses and figures were produced with the libraries of GFD-DENNOU Club.

## Author contributions

H.K. analysed the data. N.S. designed and performed the experiments. M.T., T.E., and W. O. developed the numerical model. T.S., T.M.S., and G.L.H. conducted the IR2 observation. All authors contributed to the scientific interpretation and discussion.

## Additional information

**Competing interests:** The authors declare no competing interests.

