## [Peer Review File · Nature Communications]

Reviewer #1 (Remarks to the Author):

SUMMARY:

Authors claim that they have reproduced a white long fine (streak) structure observed by Akatsuki's IR2 camera by a simplified version of AFES general circulation model. They relate this streak-structure with downward flow generated by the horizontal convergence of the meridional wind that is reinforced by baroclinic eddies generated in the low-stability layer. These baroclinic disturbance transports heat and angular momentum polewards maintaining the meridional profile of the angular velocity, which is responsible for the tilting of the pressure anomaly associated to an equatorial Rossby-like wave and, therefore, for the extended shape of the region where meridional wind converges.

They check the role of the solar heating and low static stability layer in the formation of the streak-structure by running different experiments. Authors conclude that although solar heating is not important for the formation or appearance of the streak-structure. But they do not mention the lifetime of the structure (they only show one image of the night-side of Venus), nor do they mention how the streak structure could be modified in polar regions (latitudes not well observed in the Akatsuki's image).

On the contrary, authors find that the low static stability layer plays an important role in the formation of the structure. However, I am concerned about the altitudes analyzed. It seems to me that the altitude observed in Akatsuki's images is below the low-stability layer while the simulations are shown for altitudes that are above this layer. If it is this way, it should be mentioned in the manuscript and a detailed discussion about the possible effects of this altitude discrepancy should be developed.

I also find the use of references scarce in some sections of the manuscript and that part of the text is too strong for the qualitative character of the analysis. The possible negative effects of the considerations and simplifications done in the study need to be assessed so that the reader can evaluate if the striking similarity between the observed and reproduced structures is just a coincidence or not. I believe that it is not, but I find necessary to justify it better.

Therefore, I do not recommend the publication of the manuscript in its current version. However, I believe that the matter of study in this work is novel and of great interest to the Venusian community and that after a moderate revision it would have the necessary impact to be published in this journal.

MAYOR COMMENTS:

- In the standard run and in the ZS0 sensitivity experiment the value of 0.1K/km for the static stability is used between 55 and 60 km altitudes (declared in paragraph 5 of Introduction section). In paragraph 3 of the same section, authors state that the Akatsuki images at 2.2.6microns show the clouds at about 50 km altitude. So it means that the IR2 camera images show the clouds BELOW the low-stability layer. But then authors analyze AFES-Venus models results (Figs. 2,3,4,5) at 55, 60 and 65 km altitude, that is, WITHIN or ABOVE the low-stability layer. Even if the analysis is qualitative, I think that this issue must be commented on more thoroughly. Do you see any streak structure at 50km? If not, why do you think you don't see it? I know that the altitudes in kilometers are always approximately, but where is located this 50km altitude in terms of static stability? I think it would be helpful to show a figure with the static stability profiles of the different runs with the observed and modeled altitudes remarked on it, at least in the Methods section.

- The streak structure is formed from the polar regions down to 30 degrees. There are Venus Express images at altitudes around 45km where the polar clouds are seen. Did you compare these images with your results? Is there any similarity? If yes, it would be interesting to see it. If not, why do you think it is? Can the streak structure be modified and maybe destroyed in polar areas due to the atmospheric circulation or eddies present there? I think that these issues should be discussed much more in the manuscript, authors should give a better spatial context to the streak structure.

FURTHER COMMENTS:

Please, add page and line numbers so that the review is easier to comment.

1. Abstract

- I think the sentence "Recently, a planetary-scale streak structure has been discovered in the cumulative cloud thickness of night-side images of Venus capturing infrared of 2.26microns taken by the IR2 camera..." would be better understood as "Recently, a planetary-scale streak structure has been discovered in the cumulative cloud thickness images of the night-side of Venus captured at 2.26microns infrared radiation by the IR2 camera on board Venus Climate Orbiter/Akatsuki."

- In the sentence "... are formed in both hemispheres with THE equatorial symmetry..." I would remove that "THE".

- Please, rewrite the last sentence. In the current version it is not clear whether you mean that the streaks are caused by the interaction of two things (baroclinic instability and Rossby-like waves) or that the baroclinic instability is due to two things (low-stability layer and Rossby-like waves).

2. Introduction

2.1. Paragraph 1

- "Observations of the Venus clouds by cameras based on the ground and on-board spacecrafts WITH various wavelengths have found..." Please replace that "WITH" with AT.

- "... a polar DIPOLE in infrared region..." If I am not wrong, the dipolar configuration is just one of the morphologies that the polar vortex can have. Please, verify it.

2.2. Paragraph 2

- "... and then observations _ started." Please, remove "WERE".

2.3. Paragraph 3

- "This image captures infrared of 2.26microns radiated from the hot atmosphere..." I think this sentence would be better understood as "This image captures infrared radiation at 2.26microns emitted by the hot atmosphere..."

- How do you know that this infrared radiation is emitted from altitudes around 30 km and then partly absorbed by clouds at 50 km? Please explain it or give references.

- I find that the use of streak and streaks is a bit confusing throughout the whole manuscript. For example, here, in the sentence "This feature is enhanced and become clearer as..." authors refer to the pair of streaks, so I think that the sentence should be "THESE featureS ARE enhanced and become clearer as..." However, in the next sentence "In the northern hemisphere, for instance, the streakS extend from the north-west to south-east..." they refer to one of those streaks, the one in the north, so it should be in singular "In the northern hemisphere, for instance, the streak extends from the north-west to south-east....". Right? Please check this question throughout the manuscript.

- Last sentence: Is it 100% sure that the clouds are thinner because of the downward flow (as you state forcefully in the second half of the sentence)? or is it just one of the possibilities (as you suggest in the first half of the sentence)? I suggest authors to change the second half of the sentence as "...the streak structure in bright colour (i.e. thin clouds) MAY IMPLY downward flows [8]."

2.4. Paragraph 4

- When detailing the resolution of the low- and high-resolution simulations as the number of grid points, please give also the equivalent resolution in degrees or kilometers.

- The initials "GCM" are not explained.

- "AFES-Venus focused on.." > focuseS on

- "...the low resolution version () of AFES-Venus has successfully reproduced superrotating zonal flow consistent with observations at the cloud level..." What about the high-resolution version? Do you use a version of the model that reproduces the superrotation or not?

- "... by the cold latitudinal band, THE SO-CALLED cold collar." I would add "the so-called" to the sentence.

2.5. Paragraph 5

- First sentence: in the first half author say ".. we succeeded in reproducing..." and then, in the second half, they say "... that has some similarity to that observed..." It seems to me that both halves are inconsistent with each other. If you reproduce it, the similarity has to be high (not some). Please

change "...HAS SOME SIMILARITY.." to something like "...IS QUITE SIMILAR..." And remove one of the "FIELD" words in the first half of the sentence.

- Zonally asymmetric diurnal variation of the solar heating: I feel like more details are needed at this point. What is the zonal dependence? What is the vertical dependence? What references do you use for the solar heating? This information is present in the Methods section, so please redirect at least the reader's attention to that section.

- Low-stability layer: see comment above.

- "... Pioneer Venus probe observationS..."

- "In the next section, we describe THE (morphological?) characteristics of the ..."

2.6. Figure 1

- Please specify in the caption the approximate altitude of the observation.

- Please add latitude (and longitude is possible) values to the grid.

- Please increase the size of the values and units of the magnitude shown in the figure.

2.7. Figure 2

- Please add latitude (and longitude is possible) values to the grid.

- It would be interesting to remark the low-stability layer in Fig.2e.

- Figs.2b, 2c, and 2d: is the colour code the same as in Fig.2e??

- Caption of the figure:

- (a) please redirect the reader to the Methods or to the text for the edge-enhanced image's details.

- (b)-(e) please specify the run, is it STD?

- (b)-(e) please give the equivalent of 3 Earth years in Venus time

- (b)-(e) panel (e) is not a horizontal cut at 60 km, please correct it.

- (b) please rewrite the sentence, it is better understood in the text

2.8. Table 1

- The values of static stability given in the table are not applied in the same altitude range according to the Methods section. Please add this information to the table and redirect the reader to the Methods section for further information about the solar heating treatment.

3. Results

3.1. Description of the streak structure

3.1.1. Paragraph 1

- “ Strong downward flows indicated by white form A streak structure similar to that observed.”

- “... is composed of several _ downward-flow narrow streaks..” Please remove “NUMBERS OF”.

- “... whereas, in the upper hemisphere, THAT > IT is slightly poleward ...” Please correct.

3.1.2. Paragraph 2

- From this point you only show AFES-Venus model’s results. You should emphasize it by adding a couple of comments in the manuscript and in the captions of the figures. For example, in the first line “Views OF THE MODELED STREAK from above the poles are shown...”

- I think that the sentence “The huge spirals in both hemispheres persistently develop mostly with equatorial symmetry; namely, they are synchronized with each other.” would be better understood as “The huge spirals in both hemispheres ARE persistently developED with MOSTLY equatorial symmetry; THAT IS, they are synchronized with each other.”

- The rotation period of the streak structure is about 6 Earth days in the model or in observations? Please specify it.

3.1.3. Paragraph 3

- "... to the altitudes of around 50 km, below which the static stability increases with _ depth. Remove "THE". Wasn't it below 55 km when the static stability started to increase with depth in the STD run??

- "The horizontal cross section of the meridional velocity field is shown in Fig.3e, ..." I would not define this figure like an horizontal cut. It is a polar or orthographic view, isn't it?

- "Consequently, at a given height level, several _ narrow streaks appear side by side..." Please remove "NUMBERS OF".

3.2. Importance of the low-stability layer

3.2.1. Paragraph 1

- "The conducted simulation runs are listed in Table 1". Please add a sentence to redirect reader's attention to the Methods section.

- How long does the streak structure last? Is it present for more than one Venus day? Is it that surprising to find that the solar heating has no effect on it? If we could observe the same altitude on the day-side of Venus, would we see the same kind of structure? In my opinion, although for a planetary scientist these questions are easy to see with the comparison of the STD and ZS0 simulations, to publish in journals like Nature Communications (whose public is more open) these issues should be mentioned, more context should be given to the object of study.

3.2.2. Paragraph 2

- "... is crucial for the FORMATION OF THE streak structure."

- "In ZS2, in which the static stability of the low-stability layer is increased from 0.1 K/km of STD to 2.0 K/km, ..." According to the Methods section, the altitude range where these values are fixed are not the same between STD and ZS2. Please, specify it. Same comment for the ZS4 run mentioned later; please give the altitude range where the value of 4K/km is applied.

- As mentioned in the mayor comments, I find that further discussion on this point is needed in the current manuscript. The 60km altitude is located just above the low-stability layer in the STD and ZS2 runs, while in ZS2 and ZS4 it is located within the “low-stability layer” (although this layer is no longer lowly stable). Should we look higher in the atmosphere in these two runs? Moreover, the altitude indicated for the Akatsuki’s observations (50km) is located below the low-stability layer in the STD and ZS2 runs, how could this affect the study? Are the observation altitude and the static stability profile used in the model incompatible? Or are you somehow suggesting that there is another low-stability layer below 50km altitude?

3.2.3. Paragraph 3

- “This trend CONFIRMS that the existence of the low-stability layer CAUSES the baroclinic instability...” This study, as stated by the authors later, is qualitative and not quantitative; recall, for example, the issue about the altitude of observation and low-stability layer. In addition, the baroclinic instability mentioned in this sentence has been obtained by numerical modeling that suggested (but not confirmed) that some of the wave activity observed in the Venus atmosphere could be of baroclinic nature. For me, the sentence is very forceful. Please, rewrite it.

- “... intense baroclinic disturbances are essential for the FORMATION OF THE streak structure.”

3.2.4. Figure 3

- Please specify the altitude level for panels from (a) to (h).
- Panels from (a) to (h) show the south pole, but is it the south pole in the model or in Venus atmosphere?
- Please add the label of the horizontal axes in panels from (i) to (l).
- Is the longitude of panel (l) right? Isn’t it 180deg?
- Please increase the size of the values and units of all the colobars.
- Caption: please give the equivalent Venus time to 30 Earth days.

3.3. Horizontal structure regulating the north-south symmetry

3.3.1. Paragraph 1

- Why are you showing -35deg, 65km and Z50 run? Please explain it.

- "...(about 75.6 m/s at the EQUATOR) shown by the white dashed line." Do the yellow and white lines show rotation periods at equator or at -35deg? If it is at equator, why? Clarify it also in the caption of Fig.4, please.

- "On 55km height, the ROTATION > PROPAGATION speed of p' is mostly THE same as that on 65km...". Please, give the exact value.

3.3.2. Figure 4

- There is no white dashed line in panel (b). Please add it.

- Caption:

- "... -35deg ON THE HEIGHTS OF (a) ..." please replace it by "... -35deg AT (a) 65 km and (b) 55 km HEIGHTS for the case..."

- Why have you changed the value for the strong downward flow from -0.05m/s to -0.03m/s?

3.3.3. Figure 5

- Please increase size of the colorbar, as well as the values on it.

- Caption:

- panels (b), (c), (f), and (g) are temporal and zonal means, what is the time interval considered for the temporal mean?

- "... (c,g) are also shown for both runs AT 65 KM HEIGHT".

3.3.4. Paragraph 2

- What do you mean with "composite mean"?

- "... propagation of THE pressure disturbance..."

- You explain what u' is, but not v' and w .

3.3.5. Paragraph 3

- "... p' is mostly the same in both _ levels as shown by Fig4." Please remove "OF THE".
- "... u' in the equatorial region on both _ levels (eastward..." Please remove "OF THE".
- "... other longitudes) shows that these waves are coupled ..."
- "... or synchronisation of disturbances in both _ hemispheres." Please remove "OF THE".

3.3.6. Paragraph 4

- I understand that the solar heating has no significant effects on the horizontal structure but why do you show results of the ZS0 run and not STD?

3.3.7. Paragraph 5

- How do you know that the vertical structure of the pressure anomalies at high-latitudes is uniform from 40 to 70km? Did you analyze it? If yes, show it please. If not, give a reference where this is mentioned.

- " from 40 to 70km, ARE due to the barotropic instability ..." Again, you are not analyzing this instability so you should not be so forceful in your statements. Please soften the sentence to "COULD BE".

3.4. Formation mechanism of the streak structure

3.4.1. Paragraph 1

- "... the plar-side halves (in _ latitudes from 45 to 65 degs) ..." Please remove THE.

- "... are tilted eastward with the increase of latitudes as shown ..." please change it to "... are tilted eastward with INCREASING LATITUDE as shown..."

- "... into the high-latitude anomalies appearing in _ latitudes around 75 degs." Please remove THE.

- When you say that the difference is associated with the intensity of the baroclinic disturbance, what difference do you refer to? The tilting of the pressure anomalies or the intensity of the downward flow? In any case, the detailed explanation is given in the next paragraph, so I think you should mention it.

3.4.2. Paragraph 2

- "... in the zonal mean flows is AN > THE appearance of the large angular velocity..."

- "... by the tilted eddies." What is tilted exactly? The baroclinic eddies or the pressure anomaly related to the Rossby-like wave?

4. Summary

- (2): "The streak structure persistently develops with the north-south symmetry regulated by THE > A zonal-wavenumber-one disturbance probably emerging from a vertical shear instability that consists of _ equatorial Rossby-like and Kelvin-like waves." Please remove THE.

- (3): "... of the polar REGIONS of the equatorial..." Would not be better to write "polar MOST SIDES"?

- (4): Again, when you say "tilting of the wave" or "tilted wave", what do you mean? the baroclinic eddies or the Rossby-like wave?

4.3. Paragraph 2

- "... simulations are dynamical ones and ..." > "... simulations are COMPLETELY dynamical and do not contain ..."

- The radiative calculations, photo-chemical reactions, and cloud processes have certainly an influence in the cloud opacity observed in the images of IR2 camera. But how can they influence your model's results?

4.4. Paragraph 3

- Even if this study is a qualitative interpretation of the streak structure, I think that the fact of being the modeled and modeled streaks located below and above the low-stability layer, respectively, is an important issue and that it should be discussed in more detail.

- How would improve your analysis by exploring a large amount of Akatsuki's observations?

- "... which is now being carried out ALONG with new implementations of physical processes."

5. Methods

5.1. Paragraph 1

- "AN > THE edge-emphasis process applied to ..."

5.2. Paragraph 2

- Please, when giving the number of grid points, give also the equivalent resolution in degrees or kilometers.

5.3. Paragraphs 3 and 4

- All the values in "days" units refer to "Earth days", right?

5.4. Paragraph 4

- "..., the relaxation time decreases almost exponentially FROM > FROM 10,000 EARTH days at the surface to 0.1 EARTH days at 120 km height."

- The horizontal structure of the solar heating on the day side is different below and above 71km altitude. Why? Any reference?

5.5. Paragraph 5

- First half of the paragraph. Why do you use this vertical distribution of the zonal wind or this initial temperature field? Any reference?

- Please give the equivalent Venus time for “2 - 4 Earth years”.

5.6. Paragraph 6

- Why do you change the altitude range of the low-stability layer? Why these altitude ranges? Any reference? A figure showing the different static stability profiles of the ZS0, ZS2, and ZS4 runs with the observation altitude and the altitudes shown in the figures of the manuscript would help a lot in understanding the importance of the low-stability layer.

Reviewer #2 (Remarks to the Author):

The novel claim of this manuscript is that 2.26-micron bright streaks recently observed by the Akatsuki mission on Venus are due to a globally coordinated, baroclinic instability. Given the high profile of the Akatsuki mission, and following the success of Venus Express, I think this result will appeal to a broad readership. The challenge, though, is to present a compelling case for the interpretation without alienating the general scientific audience by being overly technical. I think the authors have managed to strike this balance fairly well, however they may have erred on the side of removing too many of the important details.

In particular, this reader found the attribution to baroclinic and barotropic instability confusing and perhaps even conflated. I don't doubt the authors understand the mechanism operating to produce the streaks in their model, so I think this is simply a matter of adding more technical detail to the

physical interpretation given towards the end of the manuscript; I've highlighted this section in yellow with associated comments in the attached, annotated manuscript. I also suggest delving into a bit more detail on how increasing the static stability of the low-stability layer results in a weaker streak structure, particularly in model simulation ZS0. I'm thinking a simple discussion of the Rossby radius of deformation, and its relation to static stability would be sufficient, and may help organize the results into an overall understanding in terms of baroclinic instability.

Also, this reader found the global coordination of high-latitude streaks by equatorial waves to be quite compelling. But again, I'm left with a bit of confusion as to the relative roles of baroclinic instability of equatorial waves vs. those at high latitudes. Perhaps you could look at EP fluxes to distinguish the two.

Finally, this reader finds it somewhat confusing to appeal to baroclinic instability at all on such a slow-rotator as Venus, and while I realize this is a generalization of the phenomenon in which relative vorticity plays the role of planetary vorticity (coriolis), this will not be clear to a reader with some GFD background. I suggest adding some direct discussion of this to the manuscript.

Reviewer #3 (Remarks to the Author):

Review of the manuscript: "Planetary-scale streak structure reproduced in a Venus atmospheric simulation."

Authors successfully reproduce the planetary-scale streak structure observed for the first time in the Venus' night-side image taken by the IR2 camera on board Akatsuki by using a simplified high-resolution general circulation model, named AFES-Venus. The planetary-scale streak structure is reproduced as strong downward flows. Authors show that the presence of a low-stability layer – observed with radio occultation technique previously by Venus Express and more recently by Akatsuki – is indispensable for the occurrence of the streak structure. Authors suggest that baroclinic instability produced around the low stability layer is the cause of the streaks of downward flow.

General evaluation:

The Venus Climate Orbiter/Akatsuki is continuously providing us exceptional detailed images of Venus clouds that reveal complex structures and dynamics. AFES-Venus is a simplified general

circulation model that has already been able to reproduce successfully several features of the Venus circulation, such as the superrotation circulation (Sugimoto et al. 2014), and the polar atmospheric structure (Ando et al., 2016). It is remarkable that it manages to reproduce successfully in detail the planetary-scale streak structure observed for the first time by the Akatsuki IR2 camera and to provide a possible explanation of the formation mechanism.

The manuscript is well-written, logically organized, and the figures, tables and references are most of the time appropriate. This is a very good paper that deserves publication after moderate revisions are provided, some clarifications are however needed.

Detailed comments:

Abstract:

Pg 1: The IR2 camera observations of the planetary-scale streak structure are the first observations of these structures? Or were there other observations of this kind by previous missions (VMC/Venus Express; or Pioneer Venus)? If this is the first time that this kind of structure is observed in the Venus clouds, it is worth to mention it (here or in the introduction).

Pg 1: what do you mean exactly by “...cumulative cloud thickness...”?

Introduction:

Pg 3: “This image captures infrared of 2.26 μm radiated from the hot atmosphere around 30-km altitudes and partly absorbed by thick clouds lying at the altitudes of around 50 km.”

How has been determined the cloud altitude? It would be useful to add a reference here.

Pg 3: Why “...downward flows may promote evaporation of clouds...”? In the reference provided (Satoh et al. 2017) this is not really explained. Could you give more explanations or add a reference?

Pg 3: “...high-resolution simulations (480×240×120 grids)...”. I assume that 480 corresponds to the longitude, 240 to the latitude and 120 to the altitude? It would be better to specify it, here and later in the Methods section as 480(long)×240(lat)×120(height).

Results:

[Description of the streak structure]

Pg 6: "Figure 2b shows a simulated vertical velocity (w) field in STD at an altitude of 60 km..."

(See also my question on Figure 1) Why simulations are at 60 km while the IR2 image is at about 50 km? I understand that this is a qualitative comparison, but what could be the cause of this discrepancy in altitude?

Pg 6: "...whereas, in the upper hemisphere, that is slightly poleward compared..."

Is there a possible explanation for this hemispheric asymmetry in the simulations?

Pg. 6: "...namely, they are synchronised with each other."

If I understand well from what is explained later, the synchronization is not valid at latitude >75 deg? If this is the case, should be specified in the text.

Pg 6: "...about 6 Earth days, which is longer than that of the mean zonal flow at this level."

Please, specify the rotation period of the mean zonal flow at this altitude level.

Pg 6: "A sharp convergence zone..."

What do you mean here by convergence zone? Is it the region where two different flows meet and interact? Please, specify it.

[Importance of the "low-stability layer"]

Pg 7: "...the streaks in both hemispheres in ZS0 also rotate synchronously (not shown here)."

This is true at all latitudes, or only for $\text{lat} < 75$ deg? Could be interesting to add a video showing the evolution of streak structure as supplementary material.

[Horizontal structure regulating the north-south symmetry]

Pg 10: It is still not clear to me what the composite mean is and how it has been calculated.

Pg 11: "...in the north-south symmetry or synchronisation of disturbances in both of the hemispheres." This is valid only at low-latitudes (<50 deg)? Better to specify again.

Pg 11: "...and do not exhibit the north-south symmetry at least in STD and ZS0." Same question as before: why there is not north-south symmetry? What is producing the asymmetry?

[Summary]

Pg 12: "(2) The streak structure persistently develops with the north-south symmetry...". Specify that this is true for latitude <75 deg.

Pg 12: small typo: "...the angular velocity which in turn..." correct to "...the angular velocity which IS in turn...".

Table 1: Please add in the caption that the "low-stability layer" corresponds to the altitude from 55 to 60 km.

Figures: a general comment about the figures: fonts should be larger, in some cases it's almost impossible to read what is written. Also, please add a symbol/title to the colorbars.

Figure 2: Is the colorbar of Figure 1e the same for figures 1b,c,d? If yes please specify in the caption. Also, add the title to the colorbar: it would be enough the symbol w of the vertical velocity. Here the IR2 image (corresponding to about 50 km, as explained in the text) is compared to the simulation at 60 km. It should be explained in the main text the reason for this discrepancy in heights.

Figure 3: Two comments already done previously: 1) Use larger fonts for the colorbars; 2) Add a title/symbol to the colorbar to indicate that they correspond to the vertical velocity, meridional velocity and heat transport. A question regarding Figures 3e and 3f: from the simulations it appears that the meridional velocity direction may change with longitude and latitude (If I understand well, red color corresponds to meridional flow directed toward the pole, and blue flow directed toward the equator?). Has this trend (different directions of the meridional flow at the same altitude level) ever been observed?

Figure 4: Yellow lines: Could you please recheck them. If I understand well, like they are plotted now, they should correspond respectively to a period of about 3.2 Earth days (Fig a) and 7 Earth days (Fig. b). But in the main text it's written they should be 4.6 and 8.2 Earth days. Please add the symbol to the colorbar.

Figure 5: Caption: I am still not sure to understand what the composite means is. Also, specify that the arrows on the plot correspond to the vectors (u', v') . Add a symbol to the colorbar, larger fonts are needed.

General comment: In order to make the review easier, please add line numbers.

Authors' response to the reviewer#1's comments

Title: Planetary-scale streak structure reproduced in a Venus atmospheric simulation.

Authors: Hiroki Kashimura*, Norihiko Sugimoto, Masahiro Takagi, Yoshihisa Matsuda, Wataru Ohfuchi, Takeshi Enomoto, Kensuke Nakajima, Masaki Ishiwatari, Takao M. Sato, George L. Hashimoto, Takehiko Satoh, Yoshiyuki O. Takahashi, Yoshi-Yuki Hayashi

We thank the reviewer for a careful review and constructive comments. Please find below the authors' response. In this reply we denote referee's comments and questions using blue; our responses are in black and relevant text in the manuscript in brown.

First of all, we would like to note that the following parts of the manuscript were reorganized to satisfy the format requirements of Nature Communications:

- * Abstract was shortened to less than 150 words.
- * Introduction now includes a background of observation of Venus' night-side and a brief summary of our results and conclusion.
- * Since Figure cannot be referred in Introduction, description of the planetary-scale streak structure in IR2 image was moved to the Results section.
- * Summary was replaced by Discussion.

SUMMARY:

Authors claim that they have reproduced a white long fine (streak) structure observed by Akatsuki's IR2 camera by a simplified version of AFES general circulation model. They relate this streak-structure with downward flow generated by the horizontal convergence of the meridional wind that is reinforced by baroclinic eddies generated in the low-stability layer. These baroclinic disturbance transports heat and angular momentum polewards maintaining the meridional profile of the angular velocity, which is responsible for the tilting of the

pressure anomaly associated to an equatorial Rossby-like wave and, therefore, for the extended shape of the region where meridional wind converges.

They check the role of the solar heating and low static stability layer in the formation of the streak-structure by running different experiments. Authors conclude that although solar heating is not important for the formation or appearance of the streak-structure. But they do not mention the lifetime of the structure (they only show one image of the night-side of Venus), nor do they mention how the streak structure could be modified in polar regions (latitudes not well observed in the Akatsuki's image).

- ➡ The simulated streak structure seems to appear permanently. We obtained the streak structure at least for 3 Venusian solar days of our time-integration. This information was added in the Results section. The role of the polar disturbances, which was previously reported as the barotropic instability by Ando et al. (2017), was reanalyzed. Then we found that the polar disturbances have meridional heat fluxes and associate high-latitude jets as shown in the new Fig. 8. Because of the meridional heat fluxes, we interpreted this instability as (another) baroclinic instability. The associated high-latitude jets might contribute to the meridional tilting of the equatorial Rossby-like wave, which is a key for the formation of the streak structure. This is now discussed in the Discussion section.

On the contrary, authors find that the low static stability layer plays an important role in the formation of the structure. However, I am concerned about the altitudes analyzed. It seems to me that the altitude observed in Akatsuki's images is below the low-stability layer while the simulations are shown for altitudes that are above this layer. If it is this way, it should be mentioned in the manuscript and a detailed discussion about the possible effects of this altitude discrepancy should be developed.

- ➡ Thank you for pointing out this. We added a discussion on this topic in the new Discussion section. More detailed explanation is written in the reply to the major comments below.

I also find the use of references scarce in some sections of the manuscript and that part of the text is too strong for the qualitative character of the analysis. The possible negative effects of the considerations and simplifications done in the study need to be assessed so that the reader can evaluate if the striking similarity

between the observed and reproduced structures is just a coincidence or not. I believe that it is not, but I find necessary to justify it better.

- ➔ We added 23 references and the total number of the references is now 48 including 4 Methods-only-references. The strong expressions were modified as suggested. We have carefully considered possible negative effects; however, we found that it is difficult to provide a meaningful discussion even qualitatively, because there are too many unknown aspects in the Venusian atmospheric physical/chemical (non-dynamical) processes. (Note that, to the best of our knowledge, Venusian GCMs with a sophisticated radiative transfer code still have some difficulties in reproducing the realistic superrotation; for example, they require some artificial tuning of distribution of clouds and haze.) Hence, we just provided a comment and a possible example of physical processes that can affect the circulation in the Discussion section.

Therefore, I do not recommend the publication of the manuscript in its current version. However, I believe that the matter of study in this work is novel and of great interest to the Venusian community and that after a moderate revision it would have the necessary impact to be published in this journal.

MAJYOR COMMENTS:

- In the standard run and in the ZS0 sensitivity experiment the value of 0.1K/km for the static stability is used between 55 and 60 km altitudes (declared in paragraph 5 of Introduction section). In paragraph 3 of the same section, authors state that the Akatsuki images at 2.2.6microns show the clouds at about 50 km altitude. So it means that the IR2 camera images show the clouds BELOW the low-stability layer. But then authors analyze AFES-Venus models results (Figs. 2,3,4,5) at 55, 60 and 65 km altitude, that is, WITHIN or ABOVE the low-stability layer. Even if the analysis is qualitative, I think that this issue must be commented on more thoroughly. Do you see any streak structure at 50km? If not, why do you think you don't see it? I know that the altitudes in kilometers are always approximately, but where is located this 50km altitude in terms of static stability? I think it would be helpful to show a figure with the static stability profiles of the different runs with the observed and modeled altitudes remarked on it, at least in the Methods section.

➔ First of all, the usage of “low-stability layer” was not good in the previous manuscript. The altitudes of 55–60 km was the layers low-stability *in the basic temperature profile for Newtonian cooling*, but the static stability profile of *the simulated temperature field* shows that the low-stability layer is formed in around 50–60 km height as shown in the new Fig. 3 (simulated static-stability profile). We changed the usage of “low-stability layer” to refer to the simulated low-stability layer (i.e., ~50–60 km) throughout the manuscript.

Next, we rechecked the altitude for the IR2 night-side image and found that the description of IR2 night-side altitude in the previous manuscript was too rough. The recent study by Haus et al. (2013) proposed that the clouds particles with size of about 3.7 μm , to which the 2.26 μm wavelength (IR2 night-side image) is sensitive, are distributed in 49–57 km altitudes. This point was also corrected in the manuscript and a background for the observation of the lower clouds using the spectral window of the CO₂ atmosphere in Introduction.

Finally, even with the above two corrections, the discrepancy in the observed altitude and analyzed one exists: the streak structure in our simulation is clearer at 60 (65) km height, which is above the low-stability and middle- and lower-cloud layers. However, considering that a penetration depth (vertical displacement) due to the strong downward flow reaches about 15 km, the downward flow can significantly affect the opacity of the middle- and lower-clouds. This is mentioned in Result section (end of the first paragraph) and a detailed discussion is given in the new Discussion section.

- The streak structure is formed from the polar regions down to 30 degrees. There are Venus Express images at altitudes around 45km where the polar clouds are seen. Did you compare these images with your results? Is there any similarity? If yes, it would be interesting to see it. If not, why do you think it is? Can the streak structure be modified and maybe destroyed in polar areas due to the atmospheric circulation or eddies present there? I think that these issues should be discussed much more in the manuscript, authors should give a better spatial context to the streak structure.

➔ Thank you for this comment. Compared with VIRTIS polar images (e.g., Garate-Lopez *et al.*, 2013), a similar dipole feature was obtained in our simulations as shown in Fig. 2c,d. Analyses of the polar region in AFES-Venus simulations including polar cold collar, vortex, or dipole structure were

reported and discussed in Ando *et al.* (2016). Hence, we avoid repeating them in this paper and just mention the similarity and the previous work.

We agree that the influence of the polar region (vortex) to the streak structure should be discussed. Also, we reconsidered the latitudinal range of the streak structure and we recognized that we should not include the polar region for the streak structure, because the equatorial Rossby-like wave, which is a key for the formation of the streaks, does not reach the polar region. In other words, the polar disturbances, could prevent the Rossby-like wave from reaching the polar regions. In addition, through the revision, we recognized that the polar disturbances also transport heat poleward as shown in the new Fig. 8, so that we interpret these disturbances as baroclinic instability that may contribute to the formation of the streak structure by inducing high-latitude jets by PV stirring.

The related changes in the manuscript are as follows:

L.130-134: “Polar projections of simulated streaks are shown in Figs. 2c and 2d. In the polar regions (about > 75 deg.) of each hemisphere, there is a dipole structure similar to the polar vortex observed by VIRTIS¹⁰, as previously reported³⁰. The streaks extend from the edge of the polar region and form huge spirals in each hemisphere; the morphology is similar to the spiral observed in the hemispheric image obtained by VIRTIS³. The simulated spirals and polar dipoles seem to be evolving independently.”

L.253-264: “There are strong signals in the polar regions (> 75 deg.). However, the north-south symmetry no longer holds there. The polar anomalies in both hemispheres develop independently from the symmetric equatorial waves at least in STD and ZS0. In other words, the polar pressure anomalies could prevent the equatorial waves and the streak structure from reaching the polar regions. We should note that, as mentioned earlier, those polar disturbances propagate faster, and they should eventually vanish in the composite fields if we took the average from the date with the far longer time period. We should also note that the structure of polar anomalies is quite deep ranging over the altitudes from 40 to 70 km locating roughly along the isentropic surface (shown in Supplementary Fig. 1). Since these polar disturbances seem to have locally uniform structure in the vertical direction, the instability generating them might be interpreted as the barotropic instability as previously reported³¹. However, because of the presence of the meridional eddy heat fluxes shown in Fig. 5, we interpret that the high-

latitude baroclinic instability that is separated from the mid-latitude baroclinic instability also contributes to the polar disturbances (see also Discussion and Fig. 8).”

L.289-305: The first paragraph of Discussion.

FURTHER COMMENTS:

Please, add page and line numbers so that the review is easier to comment.

- ➡ We sincerely apologize for lacking the line number. We added the line number in the revised manuscript.

1. Abstract

- I think the sentence “Recently, a planetary-scale streak structure has been discovered in the cumulative cloud thickness of night-side images of Venus capturing infrared of 2.26microns taken by the IR2 camera...” would be better understood as “Recently, a planetary-scale streak structure has been discovered in the cumulative cloud thickness images of the night-side of Venus captured at 2.26microns infrared radiation by the IR2 camera on board Venus Climate Orbiter/Akatsuki.”

- ➡ To satisfy the word limit in Abstract, we removed the expression of 2.26 microns, and shorten the sentence as follows:

L.24-26: “Recently, a planetary-scale streak structure has been discovered in middle- and lower-cloud images of Venus’ night-side taken by the IR2 camera on board the Akatsuki orbiter.”

- In the sentence “... are formed in both hemispheres with THE equatorial symmetry...” I would remove that “THE”.

- ➡ We removed THE as suggested. (L.29)

- Please, rewrite the last sentence. In the current version it is not clear whether you mean that the streaks are caused by the interaction of two things (baroclinic instability and Rossby-like waves) or that the baroclinic instability is due to two things (low-stability layer and Rossby-like waves).

- ➡ We revised the sentence as follows: (the sentence was shortened for the word limit)

L. 31-33: “The streaks of downward flow result from meridional tilting of the Rossby-like waves’ phase lines associated with the characteristics of baroclinic instability produced around the low-stability layer.”

2. Introduction

2.1. Paragraph 1

- “Observations of the Venus clouds by cameras based on the ground and on-board spacecrafts WITH various wavelengths have found...” Pleaser replace that “WITH” with AT.

➡ We replaced WITH with AT as suggested. (L.38)

- “... a polar DIPOLE in infrared region...” If I am not wrong, the dipolar configuration is just one of the morphologies that the polar vortex can have. Please, verify it.

➡ We agree with the reviewer. We should have written as “dipolar polar vortex”. However, we reorganized and rewrote Introduction to add a background and previous studies of infrared observations of Venus. And the mention to the dipolar polar vortex was removed.

2.2. Paragraph 2

- “... and then observations _started.” Please, remove “WERE”.

➡ This sentence was removed due to the reorganization of Introduction.

2.3. Paragraph 3

- “This image captures infrared of 2.26microns radiated from the hot atmosphere...” I think this sentence would be better understood as “This image captures infrared radiation at 2.26microns emitted by the hot atmosphere...”

➡ This sentence was removed due to the reorganization of Introduction.

- How do you know that this infrared radiation is emitted from altitudes around 30 km and then partly absorbed by clouds at 50 km? Please explain it or give references.

➡ As described above in the reply to the major comment, we rechecked the altitude for the IR2 night-side image and revised Introduction to describe the background of observation in 2.26 μm wavelength.

- I find that the use of streak and streaks is a bit confusing throughout the whole manuscript. For example, here, in the sentence “This feature is enhanced and become clearer as...” authors refer to the pair of streaks, so I think that the sentence should be “THESE featureS ARE enhanced and become clearer as...” However, in the next sentence “In the northern hemisphere, for instance, the streakS extend from the north-west to south-east....” they refer to one of those streaks, the one in the north, so it should be in singular “In the northern

hemisphere, for instance, the streak extends from the north-west to south-east....". Right? Please check this question throughout the manuscript.

- ➔ In this study, we use "streaks" to refer to many white narrow lines seen in the IR2 image and our simulation; and we use "streak structure" to refer to a structure that bundles of many "streaks" are located in both hemispheres with a rough equatorial symmetry. So that, in most cases, we use "streaks" in a plural form and "streak structure" in a singular form. In order to make this clearer to readers, we modified the sentence as follows:

L. 106-109: "One of the most prominent features in this image is the existence of a planetary-scale "streak structure" that is composed of many bright streaks located in both hemispheres with a rough equatorial symmetry. The streaks are enhanced and become clearer as shown in Fig. 2a by an edge-emphasis process described in Methods."

- Last sentence: Is it 100% sure that the clouds are thinner because of the downward flow (as you state forcefully in the second half of the sentence)? or is it just one of the possibilities (as you suggest in the first half of the sentence)? I suggest authors to change the second half of the sentence as "...the streak structure in bright colour (i.e. thin clouds) MAY IMPLY downward flows [8]."

- ➔ There could be other possibilities for thinner clouds, so that we added MAY before IMPLY as suggested. (L.112)

2.4. Paragraph 4

- When detailing the resolution of the low- and high-resolution simulations as the number of grid points, please give also the equivalent resolution in degrees or kilometers.

- ➔ We removed "(480x240x120 grids)" and added a new sentence as follow:

L.78-80: "The number of grid points are 480×240 in the longitude×latitude coordinate with about 0.75-degree intervals and 120 in vertical with about 1-km intervals."

For the expression of low-resolution, the sentences were changed. In the revised texts we used the grid intervals for the description.

- The initials "GCM" are not explained.

- ➔ GCM is now defined in the following sentence:

L.61-63: “The numerical model used in this study is a simplified atmospheric general circulation model (GCM) named AFES-Venus^{20,21}, which is a Venus version of AFES, the Atmospheric GCM for the Earth Simulator^{22,23}.”

- “AFES-Venus focused on..” > focuseS on

➡ This phrase was removed due to the reorganization of Introduction.

- ...”the low resolution version () of AFES-Venus has successfully reproduced superrotating zonal flow consistent with observations at the cloud level...” What about the high-resolution version? Do you use a version of the model that reproduces the superrotation or not?

➡ The high-resolution version used in this study also reproduces Venusian superrotational flow at least for four Earth years of time-integration. Introduction section was reorganized and the related sentences were rewritten.

- “... by the cold latitudinal band, THE SO-CALLED cold collar.” I would add “the so-called” to the sentence.

➡ Another reviewer suggested to replace “;” to “or”, so that we rewrote this text as follows:

L.72: “the cold latitudinal band or so-called cold collar”

2.5. Paragraph 5

- First sentence: in the first half author say “.. we succeeded in reproducing...” and then, in the second half, they say “.. that has some similarity to that observed...” It seems to me that both halves are inconsistent with each other. If you reproduce it, the similarity has to be high (not some). Please change “...HAS SOME SIMILARITY..” to something like “...IS QUITE SIMILAR...” And remove one of the “FIELD” words in the first half of the sentence.

➡ As suggested, we modified the sentence as follows:

L.76-77: “Here, we succeeded, for the first time, in reproducing the planetary-scale streak structure in the vertical velocity field that is quite similar to that observed in the night-side image of IR2.”

- Zonally asymmetric diurnal variation of the solar heating: I feel like more details are needed at this point. What is the zonal dependence? What is the vertical dependence? What references do you use for the solar heating? This information is present in the Methods section, so please redirect at least the reader's attention to that section.

- Low-stability layer: see comment above.

➡ We added “see the Methods section for details of the experimental settings.” in the following sentence.

In the revised manuscript, explaining the detail of settings here is difficult for the word limit.

L.80-81: “Four simulation runs were conducted: one is a standard run (STD) and the others are sensitivity experiments (ZS0, ZS2, and ZS4; see Methods for details of the experimental settings).”

- “... Pioneer Venus probe observationS...”

➡ We modified the word as suggested. (L.84)

- “In the next section, we describe THE (morphological?) characteristics of the ...”

➡ This sentence was removed.

2.6. Figure 1

- Please specify in the caption the approximate altitude of the observation.

- Please add latitude (and longitude if possible) values to the grid.

- Please increase the size of the values and units of the magnitude shown in the figure.

➡ We modified the figure and caption as the reviewer requested above.

2.7. Figure 2

- Please add latitude (and longitude if possible) values to the grid.

➡ We added the values of latitude and longitude.

- It would be interesting to remark the low-stability layer in Fig.2e.

➡ We added horizontal dotted lines to indicate 50 and 60 km heights which are approximate boundaries of the simulated low-stability layer.

- Figs.2b, 2c, and 2d: is the colour code the same as in Fig.2e??

➡ Yes. We modified the caption to mention this.

- Caption of the figure:

- (a) please redirect the reader to the Methods or to the text for the edge-enhanced image’s details.

➡ We added “see Methods for the edge-emphasis process” at the end of the first sentence.

- (b)-(e) please specify the run, is it STD?

➡ Yes. We added it in the caption.

- (b)-(e) please give the equivalent of 3 Earth years in Venus time
 - ➔ We added “(about 9.38 Venusian solar days)” in the caption.

- (b)-(e) panel (e) is not a horizontal cut at 60 km, please correct it.

- (b) please rewrite the sentence, it is better understood in the text

- ➔ We rewrote the second sentence in the caption to meet the above two requests as follows:

L.574-579: “(b) Vertical velocity at an altitude of 60 km simulated in STD after 3 Earth years (about 9.38 Venusian solar days) of time-integration, shown as a satellite view mimicking the panel (a). (c) and (d) Same as (b) but for orthographic views from the south pole and the north pole, respectively, at the same altitude. (e) Same as (b) but for meridional cross-section at lon = 60 deg.; horizontal dotted lines indicate 50 and 60 km heights, which are the approximate boundaries of the simulated low-stability layer.”

2.8. Table 1

- The values of static stability given in the table are not applied in the same altitude range according to the Methods section. Please add this information to the table and redirect the reader to the Methods section for further information about the solar heating treatment.

- ➔ We added the altitude range as suggested. Also, we modified the caption to redirect the reader to Methods, as follows:

L.623-626: “Table 1: Experimental settings of the simulation runs. The right two columns show the static stability and the altitude range of the “low-stability layer” given in the basic temperature profile for Newtonian cooling shown by dashed lines in Fig. 3. See Methods for the details of the solar heating and the “low-stability layer”.”

3. Results

3.1. Description of the streak structure

3.1.1. Paragraph 1

- “ Strong downward flows indicated by white form A streak structure similar to that observed.”

➡ We added “a” as suggested. (L.118)

- “... is composed of several _ downward-flow narrow streaks..” Please remove “NUMBERS OF”.

➡ We removed “numbers of” as suggested.

- “... whereas, in the upper hemisphere, THAT > IT is slightly poleward ...” Please correct.

➡ We corrected this as

L.121-122: “in the upper hemisphere, the simulated streaks are located slightly poleward”
following another reviewer’s suggestion.

3.1.2. Paragraph 2

- From this point you only show AFES-Venus model’s results. You should emphasize it by adding a couple of comments in the manuscript and in the captions of the figures. For example, in the first line “Views OF THE MODELED STREAK from above the poles are shown...”

➡ We changed the sentence as

L.128: “Polar projections of simulated streaks are shown in Figs. 2c and d.”

following another reviewer’s suggestion. The figure caption was changed as described above.

- I think that the sentence “The huge spirals in both hemispheres persistently develop mostly with equatorial symmetry; namely, they are synchronized with each other.” would be better understood as “The huge spirals in both hemispheres ARE persistently developed with MOSTLY equatorial symmetry; THAT IS, they are synchronized with each other.”

➡ We rewrote the sentence as follows:

L.134-135: “The simulated spirals in both hemispheres are persistently developed with mostly equatorial symmetry (i.e., they are synchronised with each other),”

- The rotation period of the streak structure is about 6 Earth days in the model or in observations? Please specify it.

➡ This is in the model. We added “simulated” before “streak structure”. (L.137)

3.1.3. Paragraph 3

- “... to the altitudes of around 50 km, below which the static stability increases with _ depth. Remove “THE”. Wasn’t it below 55 km when the static stability started to increase with depth in the STD run??

- ➔ It is true that static stability in the basic state for the Newtonian cooling starts increasing with depth from 55 km; however, the static stability in the simulated atmosphere in the model run starts increasing from about 52 km. Here, an important point is that downward flow seems to penetrate until the static stability becomes high. We modified the sentence as follows:

L.142-144: “A snapshot of the meridional cross-section of vertical velocity (Fig. 2e) shows that downward flows of the streaks penetrates to the altitudes of around 50 km, where the mean static stability becomes as high as 1.2 K km^{-1} as shown in Fig. 3a.”

- “The horizontal cross section of the meridional velocity field is shown in Fig.3e, ...” I would not define this figure like an horizontal cut. It is a polar or orthographic view, isn't it?

- ➔ We changed the word “horizontal cross section” to “polar view”. (L.145)

- “Consequently, at a given height level, several _ narrow streaks appear side by side...” Please remove “NUMBERS OF”.

- ➔ We removed the words as suggested.

3.2. Importance of the low-stability layer

3.2.1. Paragraph 1

- “The conducted simulation runs are listed in Table 1”. Please add a sentence to redirect reader's attention to the Methods section.

- ➔ We changed the sentence as follows:

L.155-156: “The conducted simulation runs are listed in Table 1 and their details are described in Methods.”

- How long does the streak structure last? Is it present for more than one Venus day? Is it that surprising to find that the solar heating has no effect on it? If we could observe the same altitude on the day-side of Venus, would we see the same kind of structure? In my opinion, although for a planetary scientist these questions are easy to see with the comparison of the STD and ZS0 simulations, to publish in journals like Nature Communications (whose public is more open) these issues should be mentioned, more context should be given to the object of study.

➔ In our simulation, the streak structure was permanently obtained at least for 1 Earth year (about 3 Venus solar day). It is surprising that the diurnal variation of the solar heating has no significant effects on the streak structure because the solar diurnal variation seems to be the largest equatorially symmetric external forcing. (Note that the zonally symmetric component of the solar heating, which causes meridional temperature difference, is of course important.) Our experimental results suggest that we can expect to observe the similar streak structure on the day-side of Venus when it is not on the night-side.

Thank you for the suggestion for mentioning about this. We added some sentences at the end of this paragraph to mention the surprise of the unimportance of diurnal solar heating variation as follows:

L.160-166: “These results indicate that the diurnal variation of the solar heating does not play an essential role in the formation of the streak structure or their inter-hemispheric synchronisation. This may be surprising because the solar diurnal variation seems to be the largest, equatorially symmetric external forcing (at least in the altitudes of 50–70 km) in the Venus atmosphere. The inter-hemispheric synchronisation of the planetary-scale streak structure in ZS0 suggests that the streak structure exists in the Venus atmosphere irrespective of the local time through the Venusian solar day.”

The information of the lifetime in our simulation was added as follows:

L.139-140: “The streak structures appear permanently in our simulation at least for 1 Earth year (about 3 Venusian solar days).”

Although, our results suggest the persistent existence of the streak structure, it has not been observationally confirmed yet. The idealised settings in our simulation (including the omission of topography) may result in the unrealistically long lifetime of the streak structure or omitted physical processes might conceal the streak structure. Observational study on the lifetime of the streak structure is an important future work. This is shortly mentioned in the Discussion section as follows:

L.360-363: “The persistence of the streak structure is also strong in our simulation, while it has not been confirmed by observations. The streak structure in Venus could be less persistent because intermittent disturbances such as topographic gravity waves might disturb or conceal the structure. Effects of realistic geometry and topography on the streak structure need to be explored.”

3.2.2. Paragraph 2

- “... is crucial for the FORMATION OF THE streak structure.”

➡ Thank you for the suggestion; however, we considered “appearance” would be more appropriate in this context. We modified the sentence as follows:

L.168-169: “In contrast to the solar diurnal heating, the low-stability layer is crucial for the appearance of the streak structure.”

- “In ZS2, in which the static stability of the low-stability layer is increased from 0.1 K/km of STD to 2.0 K/km, ...” According to the Methods section, the altitude range where these values are fixed are not the same between STD and ZS2. Please, specify it. Same comment for the ZS4 run mentioned later; please give the altitude range where the value of 4K/km is applied.

➡ In the revised manuscript, the altitude range for each experiment is shown in the new Fig. 3 (and also in Table 1), so that, now, we consider it is enough to lead readers to Fig. 3. Here, we added “The profiles of the given and resultant static stability are shown in Fig. 3” (L.171).

- As mentioned in the major comments, I find that further discussion on this point is needed in the current manuscript. The 60km altitude is located just above the low-stability layer in the STD and ZS2 runs, while in ZS2 and ZS4 it is located within the “low-stability layer” (although this layer is no longer lowly stable). Should we look higher in the atmosphere in these two runs? Moreover, the altitude indicated for the Akatsuki’s observations (50km) is located below the low-stability layer in the STD and ZS2 runs, how could this affect the study? Are the observation altitude and the static stability profile used in the model incompatible? Or are you somehow suggesting that there is another low-stability layer below 50km altitude?

➡ It is true that the discrepancy in observed and simulated altitudes exists. As described in the reply to the major comments, we added a figure of the simulated static-stability profile, we revised the usage of “low-stability layer” and observation altitude of IR2, and we added a discussion of their relation.

3.2.3. Paragraph 3

- “This trend CONFIRMS that the existence of the low-stability layer CAUSES the baroclinic instability...” This study, as stated by the authors later, is qualitative and not quantitative; recall, for example, the issue about the altitude of observation and low-stability layer. In addition, the baroclinic instability mentioned in this sentence has been obtained by numerical modeling that suggested (but not confirmed) that some of the wave activity observed in the Venus atmosphere could be of baroclinic nature. For me, the sentence is very forceful. Please, rewrite it.

➡ Here, we intended to say about the baroclinic instability obtained in the Venus atmospheric simulations in previous studies (Sugimoto et al. 2014ab; Lebonnois 2016). The sentence in the previous manuscript was very misleading. In the revised version, we showed the distribution of the potential vorticity and added a brief discussion to support the description of the baroclinic instability. The sentence pointed out was revised as follows:

L.204-207: “Those PV trends confirm that the existence of the low-stability layer may cause the baroclinic instability as previously reported in simulation studies of Venus atmosphere^{20,21,36}, although the structure seems to be more complicated than formerly expected (see Discussion), and suggest that intense baroclinic disturbances are essential for the formation of the streak structure.”

- “... intense baroclinic disturbances are essential for the FORMATION OF THE streak structure.”

➡ We added the words as suggested. (L.207)

3.2.4. Figure 3

- Please specify the altitude level for panels from (a) to (h).

➡ These all are at 60 km height, and we added it in the caption. (L.589)

- Panels from (a) to (h) show the south pole, but is it the south pole in the model or in Venus atmosphere?

➡ It is the south pole in the model (whose rotation direction is the same as Earth’s one). We added “in the model” after “the south pole” in the caption. (L.590)

- Please add the label of the horizontal axes in panels from (i) to (l).

➡ We added the label of the horizontal axes for panels from (i) to (l).

- Is the longitude of panel (l) right? Isn’t it 180deg?

➡ The reviewer is right. Thank you very much for pointing out. We modified the longitude.

- Please increase the size of the values and units of all the colobars.

➡ We enlarged the fonts through the figure.

- Caption: please give the equivalent Venus time to 30 Earth days.

➡ Here, we considered giving the Venusian solar day is meaningless because the main focus is on the comparison among experiments with zonally symmetric heating (Fig. 5b, c, d), so that we added an

expression of “about 5–7 periods of the superrotation at 60 km height” as the equivalent Venus time of 30 Earth days.

3.3. Horizontal structure regulating the north-south symmetry

3.3.1. Paragraph 1

- Why are you showing -35deg, 65km and ZS0 run? Please explain it.

➡ We added a sentence after the first one to explain the reason.

L.211-214: “Here, assuming that the streak structures in STD and ZS0 are caused by the same mechanism, we analyse ZS0, which does not have the influence of the diurnal heating, to investigate the reason for the north-south symmetry of the structure. The latitude and the height used for Fig. 6a are where the amplitude of an equatorial Rossby-like wave explained below is large in this case.”

- “...(about 75.6 m/s at the EQUATOR) shown by the white dashed line.” Do they yellow and white lines show rotation periods at equator or at -35deg? If it is at equator, why? Clarify it also in the caption of Fig.4, please.

➡ Thank you very much for pointing out this problem. We noticed that the description was confused and misleading; yellow and white lines were for -35 deg. while values in the text were for the equator. We revised the rotation period and speed at the equator to those at lat = -35 deg in the text. (L.216-218)

- “On 55km height, the ROTATION > PROPAGATION speed of p' is mostly THE same as that on 65km...”. Please, give the exact value.

➡ We carefully considered the reviewer’s comments. Strictly speaking the propagation speed of the pressure anomaly slightly varies with time and height. In addition, the variation is slightly larger at 55 km, so that giving the exact value (and the white dashed line, see the next reply) may mislead readers’ focus to an unimportant detail. Hence, we hesitate for giving the exact value at 55 km.

We modified the sentence as follows:

L.221-222: “We can recognise a clear signal in p' , though it is noisy compared to the upper level, which propagates mostly in the same speed as that at 65 km.”

3.3.2. Figure 4

- There is no white dashed line in panel (b). Please add it.

➔ We reconsidered about the white dashed line and determined to remove it. As stated in the previous reply, the propagation speed of the pressure anomaly slightly varies with time and height. Giving a white dashed line could lead the readers to focus on this slight difference in the propagation speed, which is not important or focused on. To avoid such misleading, we remove the white dashed line from the figure. We believe the propagation of the pressure anomaly is obvious and the indicating line is not needed.

- Caption:

- "... -35deg ON THE HEIGHTS OF (a) ..." please replace it by "... -35deg AT (a) 65 km and (b) 55 km HEIGHTS for the case..."

➔ We changed the text as suggested. (L.601)

- Why have you changed the value for the strong downward flow from -0.05m/s to -0.03m/s?

➔ It is just a problem of the graphical appearance. If -0.05 m/s is used for the new Fig. 6, green hatches almost disappear from the panel (b) and the propagation of w cannot be detected. If -0.03 m/s is used for the new Fig. 4, the streaky structure becomes fuzzy.

3.3.3. Figure 5

- Please increase size of the colorbar, as well as the values on it.

➔ We enlarged the size of the color bar and the values in the figure.

- Caption:

- panels (b), (c), (f), and (g) are temporal and zonal means, what is the time interval considered for the temporal mean?

➔ The time interval (averaging period) is 30 Earth days same as the new Fig. 5. We added the following sentence:

L.611-612: "The period of time-average is the same as that in Fig. 5."

- "... (c,g) are also shown for both runs AT 65 KM HEIGHT".

➔ This part of the caption was rewritten as follows:

L.609-613: "(b) and (f) Temporal and zonal mean angular velocity (solid line with shading, scales written in bottom axis) and zonal wind (thick dashed line, top axis) at 65 km height for ZS0 and ZS4,

respectively. The period of time-average is the same as that in Fig. 5. (c) and (g) Same as (b) and (f) but for meridional transports of angular momentum by eddies.”

3.3.4. Paragraph 2

- What do you mean with “composite mean”?

- ➔ In the field of meteorology, composite mean is a kind of time-averaging method to highlight a propagating structure. Here, we time-averaged the snapshots of the horizontal maps that are zonally shifted according to the propagation speed and time. To express the composite mean, we modified the sentence as follows:

L.226-229: “In order to investigate the disturbance structure, we took composite means of pressure disturbance p' , horizontal wind disturbance (u', v') and vertical flow w following the propagation of the dominant pressure signal at 65km height shown in Fig. 6a. We can highlight the coherent structure associated with the dominant signal by averaging the snapshots of the fields shifted zonally according to its propagation.”

- “.. propagation of THE pressure disturbance...”

- ➔ We added THE as suggested. (L.227)

- You explain what u' is, but not v' and w .

- ➔ We have explained the vertical velocity w in the first sentence of the second paragraph in the Result section and the meridional velocity v in the fourth paragraph of the same section. The notation of prime was explained in the third paragraph of the **Importance of the low-stability layer** subsection. Here, we have avoided explaining v' and w again; however, if repeating them is useful for readers, we will rewrite to repeat them.

3.3.5. Paragraph 3

- “... p' is mostly the same in both _ levels as shown by Fig4.” Please remove “OF THE”.

- “... u' in the equatorial region on both _ levels (eastward...” Please remove “OF THE”.

- “... other longitudes) shows that thEse waves are coupled ...”

- “... or synchronisation of disturbances in both _ hemispheres.” Please remove “OF THE”.

- ➔ We have followed the above four suggestion. (L.240, 243, 244, 246)

3.3.6. Paragraph 4

- I understand that the solar heating has no significant effects on the horizontal structure but why do you show results of the ZS0 run and not STD?

- ➔ This is because we want to show our analyses and results to explain the north-south symmetry in a simpler situation. For example, if we plot the Hovmöller diagram (Fig. 6) with the STD case, the thermal tide is overlaid and it becomes somewhat harder to show the propagation of the pressure anomaly due to equatorial Rossby-like wave.

By following the reviewer's first comments in this subsection, we explained the reason for using ZS0.

Hence, we now consider further additional explanation is not necessary.

3.3.7. Paragraph 5

- How do you know that the vertical structure of the pressure anomalies at high-latitudes is uniform from 40 to 70km? Did you analyze it? If yes, show it please. If not, give a reference where this is mentioned.

- ➔ We confirmed our simulation data. We added a supplemental figure showing lat-height cross-section at lon = 150 and 330 deg. for the composite mean in the ZS0 case.

- "from 40 to 70km, ARE due to the barotropic instability ..." Again, you are not analyzing this instability so you should not be so forceful in your statements. Please soften the sentence to "COULD BE".

- ➔ The related sentences were much revised and we believe the sentences were softened. Please see the last paragraph of the subsection **Horizontal structure regulating the north-south symmetry.** (L.253-264)

3.4. Formation mechanism of the streak structure

3.4.1. Paragraph 1

- "... the polar-side halves (in _ latitudes from 45 to 65 degs) ..." Please remove THE.

- "... are tilted eastward with the increase of latitudes as shown ..." please change it to "... are tilted eastward with INCREASING LATITUDE as shown..."

- "... into the high-latitude anomalies appearing in _ latitudes around 75 degs." Please remove THE.

- ➔ We followed the above 3 suggestions. (L.268, 270, 271)

- When you say that the difference is associated with the intensity of the baroclinic disturbance, what difference do you refer to? The tilting of the pressure anomalies or the intensity of the downward flow? In any case, the detailed explanation is given in the next paragraph, so I think you should mention it.

➡ We revised this subsection and this sentence was removed. A mention to the baroclinic disturbances is now written at the end of this subsection, after giving detailed results.

3.4.2. Paragraph 2

- "... in the zonal mean flows is AN > THE appearance of the large angular velocity..."

➡ This phrase was removed.

- "... by the tilted eddies." What is tilted exactly? The baroclinic eddies or the pressure anomaly related to the Rossby-like wave?

➡ The pressure anomaly (p') and eddy components of the horizontal velocity (u', v'), which is related to the Rossby-like wave, are tilted. In this sentence, the transport of angular momentum by tilted eddies is explained, so that the eddies are the eddy components of the horizontal velocity (u', v').

Now, we have noticed that we had assumed that the readers know the eddy component of the horizontal wind flows along the pressure anomalies (i.e., contours of p' can be considered as streamlines) in the Rossby wave. However, for the broad potential readers in Nature Communications, it is not true. Hence, we modified and added some words in the second and third sentences in the previous paragraph as follows:

L.268-271: "In ZS0, the polar-side halves (in latitudes from 45 to 65 degs.) of the equatorial Rossby-wave-like structure in the pressure anomalies (p') and eddy circulations (u', v') are tilted eastward with increasing latitude as shown in Fig. 7a. The tilted p' and (u', v') seem to extend poleward and intrude into the polar anomalies appearing in latitudes around 75 degs."

4. Summary

Note that first paragraph of Summary in the previous manuscript was moved to end of Introduction and revised as a brief summary.

- (2): “The streak structure persistently develops with the north-south symmetry regulated by THE > A zonal-wavenumber-one disturbance probably emerging from a vertical shear instability that consists of _ equatorial Rossby-like and Kelvin-like waves.” Please remove THE.

➔ This sentence was modified as suggested. (L.91-92)

- (3): “... of the polar REGIONS of the equatorial...” Would not be better to write “polarMOST SIDES”?

➔ Thank you for the suggestion. We changed the text to use “polarmost sides”. (L.94)

- (4): Again, when you say “tilting of the wave” or “tilted wave”, what do you mean? the baroclinic eddies or the Rossby-like wave?

➔ We mean the Rossby-like wave. We added “equatorial Rossby-like” before the first “wave” in this sentence. (L.95)

4.3. Paragraph 2

- “... simulations are dynamical ones and ...” > “... simulations are COMPLETELY dynamical and do not contain ...”

➔ We added “completely” as suggested. (L.344)

- The radiative calculations, photo-chemical reactions, and cloud processes have certainly an influence in the cloud opacity observed in the images of IR2 camera. But how can they influence your model’s results?

➔ Because there are many unknown aspects in the Venusian atmospheric physical processes it is not easy to estimate the influence even qualitatively. One likely influence is a feedback from the cloud opacity to circulation through radiative heating and temperature. We added sentences to describe these things as follows:

L.344-355: “We should emphasise that our simulation are completely dynamical ones and do not contain sophisticated radiative calculation, photo-chemical reactions, or cloud processes, which should influence the cloud opacity observed in the night-side image taken by the IR2 camera. It is not easy to estimate the effects of these simplifications even qualitatively because there are many unknown aspects in the above processes in the Venus atmosphere. At least, however, the cloud opacity should give a feedback to the flows because the radiative solar heating should depend also on the cloud opacity; while the solar heating is simplified to the function of the height and solar zenith angle in our

model. The radiative calculation is also important because it may affect the stability and/or location of the low-stability layer, which is important for the formation of the streak structure; while the low-stability layer was mimicked through Newtonian cooling toward the externally given profile in our simulation. Despite such simplifications, the planetary-scale streak structure is reproduced even in the present dynamical setup. This fact suggests that the dynamical process is dominant for the formation mechanism of the streak structure in the Venus atmosphere.”

4.4. Paragraph 3

- Even if this study is a qualitative interpretation of the streak structure, I think that the fact of being the modeled and modeled streaks located below and above the low-stability layer, respectively, is an important issue and that it should be discussed in more detail.

➡ Again, as described in the reply to the major comments, we added a discussion on the relation between opacity of lower-clouds mostly located in the low-stability layer and strong downward flow located above (but penetrate into) the low-stability layer.

- How would improve your analysis by exploring a large amount of Akatsuki’s observations?

➡ We did not say “to improve”, but just “to confirm” our interpretation of the streak structure. This study is mostly numerical one and needs to be confirmed by observation. Analysis of horizontal wind estimated by cloud tracking technics is an example for the observational study.

- “... which is now being carried out ALONG with new implementations of physical processes.”

➡ We added “along” as suggested. (L.370)

5. Methods

5.1. Paragraph 1

- “AN > THE edge-emphasis process applied to ...”

➡ We modified the text as suggested. (L.375)

5.2. Paragraph 2

- Please, when giving the number of grid points, give also the equivalent resolution in degrees or kilometers.

➡ We added the grid-point intervals as follows:

L.385-386: “the number of grid points for calculating nonlinear terms is 480 (longitude) × 240 (latitude) with about 0.75-degree intervals.”

5.3. Paragraphs 3 and 4

- All the values in “days” units refer to “Earth days”, right?

➔ Right. We changed “days” to “Earth days” through the text to avoid ambiguous expression.

5.4. Paragraph 4

- “..., the relaxation time decreases almost exponentially FROM > FROM 10,000 EARTH days at the surface to 0.1 EARTH days at 120 km height.”

➔ We corrected the typo and the word “Earth” was added. (L.410-411)

- The horizontal structure of the solar heating on the day side is different below and above 71km altitude. Why?

Any reference?

➔ We followed the solar heating function proposed by Crisp (1986), who reported that the horizontal structure of the solar heating is different below and above 71 km height, because of solar radiation absorption by CO₂ above 71 km. We added some words to lead readers to the reference as follows:

L.413-415: “For the standard run (STD), the horizontal structure of the solar heating on the day side is assumed to be proportional to $\cos^{0.6}\theta_s$ above 71 km height and to $\cos^{1.5}\theta_s$ in the lower layers, where θ_s is solar zenith angle, as proposed by Crisp⁴⁶.”

5.5. Paragraph 5

- First half of the paragraph. Why do you use this vertical distribution of the zonal wind or this initial temperature field? Any reference?

➔ The vertical profile of the initial zonal wind field mimics observed zonal wind profile very simply and the initial temperature field is given to be in the cyclostrophic/centrifugal balance with the zonal wind field. We modified the sentence as follows:

L.417-418: “An idealised superrotation whose zonal wind speed reaches 100 m s^{-1} at 70 km height at the equator, very simply mimicking observed zonal-wind profiles⁴⁷, is introduced as the initial state.”

- Please give the equivalent Venus time for “2 - 4 Earth years”.

➔ We added the equivalent time in Venusian solar days as follows:

L.422-424: “After the time-integration of about two Earth years (about 6.25 Venusian solar days), the simulated atmosphere achieved a statistically steady state, and the data obtained in the fourth Earth year (after about 9.38 Venusian solar days) is used in this study.”

5.6. Paragraph 6

- Why do you change the altitude range of the low-stability layer? Why these altitude ranges? Any reference?

A figure showing the different static stability profiles of the ZS0, ZS2, and ZS4 runs with the observation altitude and the altitudes shown in the figures of the manuscript would help a lot in understanding the importance of the low-stability layer.

- ➡ Thank you very much for the suggestion. We added a new figure (Fig. 3) showing the vertical profiles of static stability in the basic temperature for Newtonian cooling and that in simulated temperature fields for each experiment. Altitudes of 60 and 50 km are also mentioned in the figure.

Authors' response to the reviewer#2's comments

Title: Planetary-scale streak structure reproduced in a Venus atmospheric simulation.

Authors: Hiroki Kashimura*, Norihiko Sugimoto, Masahiro Takagi, Yoshihisa Matsuda, Wataru Ohfuchi, Takeshi Enomoto, Kensuke Nakajima, Masaki Ishiwatari, Takao M. Sato, George L. Hashimoto, Takehiko Satoh, Yoshiyuki O. Takahashi, Yoshi-Yuki Hayashi

We thank the reviewer for a careful review and constructive comments. Please find below the authors' response. In this reply we denote referee's comments and questions using blue; our responses are in black and relevant text in the manuscript in brown.

First of all, we would like to note that the following parts of the manuscript were reorganized to satisfy the format requirements of Nature Communications:

- * Abstract was shortened to less than 150 words.
- * Introduction now includes a background of observation of Venus' night-side and a brief summary of our results and conclusion.
- * Since Figure cannot be referred in Introduction, description of the planetary-scale streak structure in IR2 image was moved to the Results section.
- * Summary was replaced by Discussion.

The novel claim of this manuscript is that 2.26-micron bright streaks recently observed by the Akatsuki mission on Venus are due to a globally coordinated, baroclinic instability. Given the high profile of the Akatsuki mission, and following the success of Venus Express, I think this result will appeal to a broad readership. The challenge, though, is to present a compelling case for the interpretation without alienating the general scientific audience by being overly technical. I think the authors have managed to strike this balance fairly well, however they may have erred on the side of removing too many of the important details.

In particular, this reader found the attribution to baroclinic and barotropic instability confusing and perhaps even conflated. I don't doubt the authors understand the mechanism operating to produce the streaks in their model, so I think this is simply a matter of adding more technical detail to the physical interpretation given towards the end of the manuscript; I've highlighted this section in yellow with associated comments in the attached, annotated manuscript. I also suggest delving into a bit more detail on how increasing the static stability of the low-stability layer results in a weaker streak structure, particularly in model simulation ZS0. I'm thinking a simple discussion of the Rossby radius of deformation, and its relation to static stability would be sufficient, and may help organize the results into an overall understanding in terms of baroclinic instability.

- ➔ Thank you for your suggestion. Discussion of the effects of the Rossby radius of deformation or static stability to the necessary condition of the baroclinic instability (Holton, 2004) seems to qualitatively agree with our experimental results. However, the situation is more complicated. In the low-stability cases, Rossby waves seem to be trapped in the layer with a sharp gradient of the static stability. On the other hand, in the high-stability cases, such a sharp gradient of the static stability was not formed, so that the Rossby waves cannot be trapped. In that case, we cannot use the Eady model of the baroclinic instability. The structure of the baroclinic instability in our simulation is also complicated possibly due to the low static stability and/or a large amplitude of the baroclinic instability. We consider a further exploration is needed to make a meaningful discussion and that should be published as a separated paper. Let us reserve this topic as our future work.

Although the structure of the baroclinic instability in our simulation is not fully explored, we believe that the instability occurs in the low-stability cases. To support this, we added a figure of the mean meridional distribution of the potential vorticity in each case (new Fig. 5e-h). From these figures, it is clear that the necessary condition of the mean PV-gradient for the baroclinic instability is not satisfied in high-stability cases (ZS2, 4) but can be satisfied in low-stability cases (STD, ZS0).

In the manuscript, we added a description of PV fields and a short discussion as follows:

L.190-207: "Increasing the static stability of the "low-stability layer" also brings a significant change in the mean (Ertel's) potential vorticity (PV) field as shown in Figs. 5e-h. In the southern hemispheres in STD and ZS0, the mean PV fields are rather complicated but have common characteristics; negative-low or even positive PV are widely distributed in the mid-latitudes of about 50-60 km heights and polar region of about 55-60 km. Here, not only the lowness of the stability but also the horizontal

component of vorticity contributes to produce positive values of PV in the southern hemisphere. In contrast to the low-stability cases, such local maxima of PV do not exist in the high-stability cases shown in Figs. 5g and 5h; note that positive PV in the equatorial region would be due to fluctuation across the equator and may disappear by taking much longer time-averaging period. As for the emergence of the baroclinic instability, a pair of layers with opposite sign in the mean meridional PV-gradient is necessary.³⁵ For in STD and ZS0, a pair of isentropic surfaces of $\theta = 830$ and 860 K and that of $\theta = 840$ and 870 K could correspond to the opposite PV-gradient layers to cause the emergence of the baroclinic instability in mid-latitudes, respectively. Similarly, another baroclinic instability could emerge near the polar positive PV region in STD and ZS0. In contrast to those low-stability cases, the PV-gradient is mostly positive especially in the mid-latitudes and does not satisfy the necessary condition for the baroclinic instability in the higher stability cases. Those PV trends confirm that the existence of the low-stability layer may cause the baroclinic instability as previously reported in simulation studies of Venus atmosphere^{20,21,36}, although the structure seems to be more complicated than formerly expected (see Discussion), and suggest that intense baroclinic disturbances are essential for the formation of the streak structure.”

L.307-318: “Though we have speculated the formation mechanism of the planetary-scale streak structure as above, we should note that the details of the disturbances and instabilities in our simulation are still unclear and remain to be explored. The situations where those disturbances are placed in the simulated Venus atmosphere, e.g., absence of surface, vertical profile of the static stability, and importance of relative vorticity playing a role of planetary vorticity in a slowly rotating planet, differ largely from those in the Earth’s troposphere. The structure of the waves and instabilities in our simulation should be quite different from those commonly considered in the Earth’s meteorology. Interactions or relations among the baroclinic instability in the high-latitudes, that in the mid-latitudes, and the shear instability in the equatorial region are also open questions. Furthermore, the mechanism for the disappearance of the intense baroclinic disturbances in the higher stability cases should be investigated. We need to understand these mechanisms to evaluate the robustness or sensitivity of the speculated formation mechanism presented here. We, however, keep the further investigations for our future studies.”

Also, this reader found the global coordination of high-latitude streaks by equatorial waves to be quite compelling. But again, I'm left with a bit of confusion as to the relative roles of baroclinic instability of equatorial waves vs. those at high latitudes. Perhaps you could look at EP fluxes to distinguish the two.

➔ As the reviewer pointed out, the roles of the baroclinic instability of equatorial waves and those at high latitudes were not distinguished. Through the revision, recognized that the polar disturbances, which we previously called barotropic instability, should be interpreted as (another) baroclinic instability because it induces poleward eddy heat fluxes and high-latitude jets as shown in Fig. 8. In our revised speculation, the role of high-latitude baroclinic instability is to form a jet by PV stirring and contribute to the meridional tilting of the equatorial Rossby wave. Please see the first paragraph of the Discussion section for the overall speculation.

We also have drawn EP-fluxes (shown below) but it seems to be less informative because $d\theta/dz$ becomes very close to zero in high-latitudes and the EP-flux becomes very large compared to those in mid-latitudes.

Fig. EP-fluxes (vectors) normalized by the planetary radius and the surface pressure and their divergence (colour) for each case. Contours show the mean potential temperature. Note that the vertical axis is the pressure coordinate and the approximate altitude range drawn is 50–70 km.

Finally, this reader finds it somewhat confusing to appeal to baroclinic instability at all on such a slow-rotator as Venus, and while I realize this is a generalization of the phenomenon in which relative vorticity plays the

role of planetary vorticity (coriolis), this will not be clear to a reader with some GFD background. I suggest adding some direct discussion of this to the manuscript.

- ➔ Thank you for the suggestion. The phenomena we called “baroclinic instability” in the manuscript satisfied the necessary condition for the baroclinic instability, so that we called it baroclinic instability. However, its structure seems to be different from the typical baroclinic instability in the Earth atmosphere. Investigation of its structure requires much more work. In this manuscript, we would like to focus on the streak structure and quickly report its reproduction in our dynamical simulation. Now, in the revised manuscript, we added the figure of mean PV distribution (new Figs. 5e-h) and mentioned the necessary condition of the baroclinic instability in the last paragraph in subsection “Importance of the low-stability layer”. How the simulated baroclinic instability seems to be complex compared to the Earth atmosphere is mentioned in the second paragraph of the Discussion section.

Comments on the manuscript PDF:

Abstract

1. “has not been investigated yet” => has not been investigated
 - ➔ We removed “yet” as suggested. (L.27)
2. “by Venus Express and Akatsuki”: There were a number of previous measurements by other missions. For instance, later you refer to Pioneer Venus.
 - ➔ We removed “Venus Express and Akatsuki” from Abstract. We would like to avoid listing all other missions in Abstract for the word limit.
3. “high-latitude strong” => strong high-latitude
 - ➔ This phrase was removed in the revision of Abstract.
4. “the” => this
5. “would be” => is
 - ➔ Above wordings (4–5) were changed as suggested.

Introduction

6. “in ultra violet region” => in the ultra violet region
7. “in infrared region” => in the infrared region

➔ Above wordings (6–7) were changed as suggested. (L.40-41)

8. “in mid-infrared region” => in the mid-infrared region

9. “the altitudes of” => altitudes of

➔ Above wordings were removed.

10. “become” => becomes

➔ We changed the sentence as follows (in the Results section):

L.108: “The streaks are enhanced and become clearer as shown in Fig. 2a...”

11. “north-west to south-east” => northwest to southeast

➔ This wording was changed as suggested. (L.110)

12. “may promote” => promote

➔ Suggested by another reviewer, we reconsidered that there could be other reason for thinner clouds, so that we keep “may” here, in addition, we added “may” before “imply”.

13. “In this study...”: This paragraph begins abruptly after the previous. I suggest adding some text to make a smoother transition.

➔ Introduction was reorganized, so that we consider the transition has become smoother.

14. “Although under” => Despite

➔ This wording was changed as suggested. (L.69)

15. “cold latitudinal band, cold collar” => cold latitudinal band or cold collar

➔ Here, another reviewer suggested to add “so-called”, so that we changed the text as follows:

“the cold latitudinal band or so-called cold collar” (L.72)

16. “one for” => one is

17. “the others for” => the others are

➔ Above wordings (16–17) were changed as suggested.

18. “the sensitivity experiments” => study of the sensitivity experiments

➔ This expression was removed.

19. “Table 1: Experimental settings of each run” => Table 1: Experimental settings of each model simulation

➔ We changed to “Table 1: Experimental settings of the simulation runs.” (L.623)

Results

[Description of the streak structure]

20. “same as” => mimicking

21. “from streak structure” => from a streak structure

➡ Above wordings (20–21) were changed as suggested.

22. “downward-flow narrow streaks” => downward-flowing narrow streaks

➡ We revised as “several narrow streaks of downward flows” (L.119)

23. “appears” => stretches

24. “latitudinal location” => the latitudinal location

➡ Above wordings (23–24) were changed as suggested.

25. “that is slightly poleward” => the simulated streak is slightly poleward

➡ We revised as “the simulated streaks are located slightly poleward” (L.122)

26. “differences exist, qualitative similarity is remarkable.” => differences from the observations exist, a qualitative similarity with the simulated streaks is notable.

➡ We revised as follows:

“There is a notable qualitative morphological similarity between the simulated and the observed streak structure, although there are quantitative differences between them.” (L.123-124)

27. “Views from above the poles” => Polar projections of simulated streaks

28. “a huge spiral” => huge spirals

➡ Above wordings (27–28) were changed as suggested.

29. “The huge spirals in both hemispheres persistently develop mostly with” => The spirals are persistent features, typically with

➡ Here, another reviewer also gave us a suggestion for rewriting and we rewrote as follows:

L.134-135: “The simulated spirals in both hemispheres are persistently developed with mostly equatorial symmetry (i.e., they are synchronised with each other),”

30. “Starting points of the downward flows are located slantwise from polar-upper region to equatorward-lower region” => Downward flows form slantwise from polar-upper regions to equatorward-lower regions

31. “shown by green hatches” => overlaid by green hatches

32. “equatorial-side edge” => upper, equatorial side edge

33. “that the sharp”: remove extra space

➡ Above wordings (30–33) were changed as suggested.

[Importance of the “low-stability layer”]

34. “filed” => field

➡ This wording was changed as suggested.

35. “in streak structure” => in streak structures

➡ We revised this part as “elongated to be recognised as a streak structure” (L.173)

[Horizontal structure regulating the north-south symmetry]

36. “at the latitude of 35 degrees south on 65 km height in ZS0” => “at 35 degrees south latitude and 65 km height in the ZS0 case”

37. “On 55 km” => At 55 km

38. “ p ’ is mostly the same” => p ’ is nearly the same

➡ Above wordings (36–38) were changed as suggested.

[Formation mechanism of the streak structure]

39. “it is strong in ZS0 and weak in ZS4 as shown in Figs. 3n and 3p”: Within the context provided in the next paragraph, please comment on how the stability of the low-stability layer affects the strength of the resulting eddy phenomena.

➡ The stability affects the strength of the resulting eddy phenomena through the emergence of the baroclinic instability or intensity of the baroclinic disturbances. In the low-stability case, the baroclinic instability emerges and induces strong disturbances; whereas, in the high-stability case, the baroclinic instability does not appear. The role of the baroclinic disturbances is described in the next reply.

40. “We speculate...”: It’s not obvious to me how the barotropic instability noted previously fits into this interpretation. Please comment and add text as necessary.

➔ Our speculation was revised. Now the speculated roles of the baroclinic instability in high-latitudes (previously called barotropic instability) and that in mid-latitudes are separately described in the Discussion section as follows:

L.289-305: “In order to consider the role of the baroclinic disturbances, locations of the heat fluxes and jets would be informative. Figure 8 shows the mean meridional distribution of the north-south eddy heat flux ($\overline{\rho v' \theta'}$) and the mean zonal wind (\bar{u}) in ZS0. There are two axes of large poleward heat fluxes, each of which implies the emergence of the baroclinic instability: one in mid-latitudes indicated by the white dashed line and the other in high-latitudes indicated by the black one, in each hemisphere. The locations of the high-latitude heat fluxes agree with those of high-latitude jets. This implies that the baroclinic instability in high-latitudes contribute to the formation of the jet by PV stirring^{40,41}, the same manner as that of the mid-latitude jets in the Earth’s troposphere. Thus, a part of the poleward eddy angular momentum fluxes shown in Fig. 7c should be caused by the stirring due to the high-latitude baroclinic instability. On the other hand, jets are not formed in the regions of the mid-latitude heat fluxes but a large amount of eddy angular-momentum fluxes exist over this region as shown in Fig. 7c and are also indicated by red arrows in Fig. 8 at the altitude of 65 km. In the mid-latitudes, the baroclinic disturbances themselves seem to be composed of the equatorial Rossby-like wave with the meridional tilting of the phase line and transporting the angular momentum poleward as shown in Fig. 7a and described in the previous section. Here, we note that the equatorial Rossby-like wave seems to compose both the baroclinic instability in the mid-latitudes and the vertical shear instability in the equatorial region at the same time. Then the north-south symmetry is regulated by the equatorial Kelvin-like wave as the other component of the latter one.”

Authors' response to the reviewer#3's comments

Title: Planetary-scale streak structure reproduced in a Venus atmospheric simulation.

Authors: Hiroki Kashimura*, Norihiko Sugimoto, Masahiro Takagi, Yoshihisa Matsuda, Wataru Ohfuchi, Takeshi Enomoto, Kensuke Nakajima, Masaki Ishiwatari, Takao M. Sato, George L. Hashimoto, Takehiko Satoh, Yoshiyuki O. Takahashi, Yoshi-Yuki Hayashi

We thank the reviewer for a careful review and constructive comments. Please find below the authors' response. In this reply we denote referee's comments and questions using blue; our responses are in black and relevant text in the manuscript in brown.

First of all, we would like to note that the following parts of the manuscript were reorganized to satisfy the format requirements of Nature Communications:

- * Abstract was shortened to less than 150 words.
- * Introduction now includes a background of observation of Venus' night-side and a brief summary of our results and conclusion.
- * Since Figure cannot be referred in Introduction, description of the planetary-scale streak structure in IR2 image was moved to the Results section.
- * Summary was replaced by Discussion.

Authors successfully reproduce the planetary-scale streak structure observed for the first time in the Venus' night-side image taken by the IR2 camera on board Akatsuki by using a simplified high-resolution general circulation model, named AFES-Venus. The planetary-scale streak structure is reproduced as strong downward flows. Authors show that the presence of a low-stability layer – observed with radio occultation technique previously by Venus Express and more recently by Akatsuki – is indispensable for the occurrence of the streak structure. Authors suggest that baroclinic instability produced around the low stability layer is the cause of the streaks of downward flow.

General evaluation:

The Venus Climate Orbiter/Akatsuki is continuously providing us exceptional detailed images of Venus clouds that reveal complex structures and dynamics. AFES-Venus is a simplified general circulation model that has already been able to reproduce successfully several features of the Venus circulation, such as the superrotation circulation (Sugimoto et al. 2014), and the polar atmospheric structure (Ando et al., 2016). It is remarkable that it manages to reproduce successfully in detail the planetary-scale streak structure observed for the first time by the Akatsuki IR2 camera and to provide a possible explanation of the formation mechanism.

The manuscript is well-written, logically organized, and the figures, tables and references are most of the time appropriate. This is a very good paper that deserves publication after moderate revisions are provided, some clarifications are however needed.

Detailed comments:

Abstract:

Pg 1: The IR2 camera observations of the planetary-scale streak structure are the first observations of these structures? Or were there other observations of this kind by previous missions (VMC/Venus Express; or Pioneer Venus)? If this is the first time that this kind of structure is observed in the Venus clouds, it is worth to mention it (here or in the introduction).

- ➔ We revised the Introduction to add a brief review of Venus night-side observation by near 1.74 μm and 2.3 μm . We mentioned previous observations including a terrestrial telescope and Galileo NIMS. A band-like structure and a spiral in Southern hemisphere were already reported but a fine streak structure in both hemispheres with north-south symmetry was not reported ever.

Pg 1: what do you mean exactly by "...cumulative cloud thickness..."?

- ➔ Our choice of the word might be inappropriate. We intended to mean the thickness of the cloud integrated from the cloud bottom to the top of the atmosphere. We reconsidered and changed the text to use "cloud opacity".

Introduction:

Pg 3: “This image captures infrared of 2.26 μm radiated from the hot atmosphere around 30-km altitudes and partly absorbed by thick clouds lying at the altitudes of around 50 km.”

How has been determined the cloud altitude? It would be useful to add a reference here.

- ➔ It was studied and discussed in Grinspoon et al. (1993). We also rechecked the observation altitude of 2.26 μm and found that the recent study by Haus (2013) proposed that the cloud altitude could be 49–57 km heights. We added these two references here. (L.43-46)

Pg 3: Why “...downward flows may promote evaporation of clouds...”? In the reference provided (Sato et al. 2017) this is not really explained. Could you give more explanations or add a reference?

- ➔ We have regretted that description on the relation between clouds and downward flow was too rough or may be incorrect. We rechecked and considered the possible mechanism for decreasing middle- and lower-cloud opacity by strong downward flow from above the middle- and lower-cloud layers. This is now discussed in the new Discussion section including the discrepancy in observed and simulated heights, as follows:

L.322-342: “Although there is a marked similarity between the planetary-scale streak structure observed in the IR2 night-side image and that represented as strong downward flow in our simulation, the altitudes of the structure differ between the observed and the simulated. The contrast features in the IR2 night-side image with 2.26 μm are considered to reflect the opacity contrast in the middle and lower clouds located around 49–57 km altitudes^{13,14}, the bottom and inside of the low-stability layer. On the other hand, the numerical model simulates the streak structure in the downward flow and the structure is clear at just above the low-stability layer ($z = 60$ km in STD and 65 km in ZS0).

As shown in Fig. 2e the downwelling flow consisting the streak structure penetrates into the low-stability layer. In addition, a penetration depth of an air parcel (H) can be roughly estimated as $H = W \times T \sim 15$ km by using $W = 0.1 \text{ m s}^{-1}$ for the typical scale of w and $T = L/U = 7.5 \times 10^6 \text{ m} / 50 \text{ m s}^{-1} = 1.5 \times 10^5 \text{ sec}$ for the timescale estimated from scales of the horizontal length (L) and propagation speed (U) of the streak structure. Hence, the vertical displacements due to the streak structure formed at above the low-stability layer would reach deep into the low-stability and middle- and lower-cloud layers.

The dynamical downward displacement in the cloud layer may decrease the middle- and lower-cloud opacity. The downward flow tends to prevent the supply of sulfuric acid vapour, which might be essential for the generation of the middle and lower clouds⁴², from the sub-cloud reservoir. Moreover, the displaced fresh air parcels include smaller cloud-particles generated at the cloud-top altitude (~ 65 km)⁴³ and push out old air which contains the larger cloud-particles, so-called “mode 3” (about 3.7 μm). These contribute to the decrease of the opacity for 2.26-μm wavelength.”

Pg 3: “...high-resolution simulations (480×240×120 grids)...”. I assume that 480 corresponds to the longitude, 240 to the latitude and 120 to the altitude? It would be better to specify it, here and later in the Methods section as 480(long)×240(lat)×120(height).

➡ Another reviewer suggested to add information of grid intervals, so that we changed as follows:

In introduction, we remove “(480×240×120 grids)”, and added a new sentence.

L.78-80: “The number of grid points is 480×240 in the longitude×latitude coordinate with about 0.75-degree intervals and 120 in vertical with about 1-km intervals.”

In Methods, we rewrote “480×240” to “480 (longitude) × 240 (latitude)”.

Results:

[Description of the streak structure]

Pg 6: “Figure 2b shows a simulated vertical velocity (w) field in STD at an altitude of 60 km...”

(See also my question on Figure 1) Why simulations are at 60 km while the IR2 image is at about 50 km? I understand that this is a qualitative comparison, but what could be the cause of this discrepancy in altitude?

➡ This discrepancy is now discussed in the Discussion section as mentioned above. The reason for the discrepancy in altitudes of streak structure would be the difference in the physical quantities (cloud opacity vs. downward flow). The strong downward flow at 60 or 65 km height in our simulation makes a vertical displacement of about 15 km, which significantly disturbs the middle- and lower-cloud layers (49–57 km).

Pg 6: “...whereas, in the upper hemisphere, that is slightly poleward compared...”

Is there a possible explanation for this hemispheric asymmetry in the simulations?

- ➔ Hemispheric asymmetry (equatorial asymmetry) is larger in the observation. Note that observer point in Figs 2a,b is not exactly above the equator but above 9.1° latitude. We added the info of sub-spacecraft point in the caption of Fig. 1 to note the reader that the observer point is not exactly above the equator.

The asymmetry in the observation would be due to some other asymmetric disturbances that are not simulated in our simplified model. In other words, our simplified settings could be an idealized situation for the streak structure.

Pg. 6: "...namely, they are synchronised with each other."

If I understand well from what is explained later, the synchronization is not valid at latitude >75 deg? If this is the case, should be specified in the text.

- ➔ The reviewer is correct. We revised the manuscript to exclude the polar region (> 75 deg) for the streak structure. The related sentences were revised as follows:

L.130-136: "In the polar regions (about > 75 deg.) of each hemisphere, there is a dipole structure similar to the polar vortex observed by VIRTIS¹⁰, as previously reported³⁰. The streaks extend from the edge of the polar region and form huge spirals in each hemisphere; the morphology is similar to the spiral observed in the hemispheric image obtained by VIRTIS³. The simulated spirals and polar dipoles seem to be evolving independently. The simulated spirals in both hemispheres are persistently developed with mostly equatorial symmetry (i.e., they are synchronised with each other), while the simulated polar dipoles are not synchronised (see Supplementary Movies 1 and 2)."

Pg 6: "...about 6 Earth days, which is longer than that of the mean zonal flow at this level."

Please, specify the rotation period of the mean zonal flow at this altitude level.

- ➔ The rotation period of the mean zonal flow is about 4.9 Earth days at the equator; we added this info as follows:

L.136-139: "The rotation period of the simulated dipoles is around 2–4 Earth days, while that of the simulated streak structure is about 6 Earth days, which is longer than that of the mean zonal flow at this level (about 4.9 Earth days at the equator)."

Pg 6: "A sharp convergence zone..."

What do you mean here by convergence zone? Is it the region where two different flows meet and interact?

Please, specify it.

- ➔ We intended to mean the region where poleward flow (blue) and equatorward flow (red) meet. Such region is indicated by the sharp boundary between blue and red with blue in the equatorward side and red in the poleward side. To specify it, we modified the sentence as follows:

L.146-148: “A sharp convergence zone of the meridional flow, shown as the boundary between equatorial-sided blue regions and polar-sided red region, is formed and its location...”

[Importance of the “low-stability layer”]

Pg 7: “...the streaks in both hemispheres in ZS0 also rotate synchronously (not shown here).”

This is true at all latitudes, or only for lat<75 deg? Could be interesting to add a video showing the evolution of streak structure as supplementary material.

- ➔ We added Supplemental Movie to show the evolution of streak structure as suggested and modified the sentence as follows:

L.158-159: “In addition, the streaks in both hemispheres in ZS0 also rotate synchronously (see Supplementary Movie 3), while polar disturbances rotate faster and independently as in STD.”

[Horizontal structure regulating the north-south symmetry]

Pg 10: It is still not clear to me what the composite mean is and how it has been calculated.

- ➔ Composite mean is a kind of averaging method to highlight a propagating structure. Here, we time-averaged the snapshots of the horizontal maps that are zonally shifted according to the propagation speed and time. To express the composite mean, we modified the sentence as follows:

L.226-229: “In order to investigate the disturbance structure, we took composite means of pressure disturbance p' , horizontal wind disturbance (u', v') and vertical flow w following the propagation of the dominant pressure signal at 65km height shown in Fig. 6a. We can highlight the coherent structure associated with the dominant signal by averaging the snapshots of the fields shifted zonally according to its propagation.”

Pg 11: “...in the north-south symmetry or synchronisation of disturbances in both of the hemispheres.” This is valid only at low-latitudes (<50 deg)? Better to specify again.

- ➔ We revised the sentences to specify the latitudinal ranges. On $z = 65$ km, Rossby-wave like structure extends to lat ~ 60 deg. and on $z = 55$ km, Kelvin-wave like structure extends to lat ~ 50 deg. We call this latitude range “the low- and mid-latitudes”.

L.229-234: “The composited structure of p' and (u', v') on 65 km height in the low- and mid-latitudes (< 60 deg.) shown in Fig. 7a resembles that of the equatorial Rossby wave³⁷ with wavenumber one; i.e., two pairs of low- and high-pressure anomalies associated with cyclonic and anti-cyclonic circulations, respectively, are straddling the equator. On the other hand, in the lower altitude at 55 km height, the composited structure in latitude < 50 deg. resembles that...”

L.244-246: “The vertical shear instability that contains the structure of the equatorial Kelvin-like wave would result in the north-south symmetry or synchronisation of disturbances in the low- and mid-latitudes in the hemispheres.”

Pg 11: “...and do not exhibit the north-south symmetry at least in STD and ZS0.” Same question as before: why there is not north-south symmetry? What is producing the asymmetry?

- ➔ We consider that the north-south symmetric disturbances due to the equatorial waves (Kelvin & Rossby) seem not to have a dominant influence in the high-latitudes, and the polar pressure anomalies in both hemispheres develop independently. Of course, by nature, barotropic instabilities (which is now interpreted as baroclinic instability because it induces large poleward eddy heat fluxes) in the polar regions not necessarily develop with north-south symmetry. We revised the manuscript as follows:

L.253-256: “There are strong signals in the polar regions (> 75 deg.). However, the north-south symmetry no longer holds there. The polar anomalies in both hemispheres develop independently from the symmetric equatorial waves at least in STD and ZS0. In other words, the polar pressure anomalies could prevent the equatorial waves and the streak structure from reaching the polar regions.”

[Summary]

Note that first paragraph of Summary in the previous manuscript was moved to the end of Introduction and revised as a brief summary.

Pg 12: “2) The streak structure persistently develops with the north-south symmetry...”. Specify that this is true for latitude < 75 deg.

➔ We modified the text as follows:

L.89-90: “The streak structure persistently develops with the north-south symmetry for low- and mid-latitudes (< 75 deg.) even without diurnal solar variation;”

Pg 12: small typo: “...the angular velocity which in turn...” correct to “...the angular velocity which IS in turn...”.

➔ The typo was corrected.

Table 1: Please add in the caption that the “low-stability layer” corresponds to the altitude from 55 to 60 km.

➔ Another reviewer suggested to add the altitude range in the Table, so that now this info is included in the Table.

Figures: a general comment about the figures: fonts should be larger, in some cases it's almost impossible to read what is written. Also, please add a symbol/title to the colorbars.

➔ We revised all figures to add symbols/titles to the colorbars and enlarge small fonts.

Figure 2: Is the colorbar of Figure 1e the same for figures 1b,c,d? If yes please specify in the caption. Also, add the title to the colorbar: it would be enough the symbol w of the vertical velocity.

➔ Yes. The color bar is common for (b)–(e). We added a note in the caption. Fonts in the figure were enlarged and the title “ w ” was added.

Here the IR2 image (corresponding to about 50 km, as explained in the text) is compared to the simulation at 60 km. It should be explained in the main text the reason for this discrepancy in heights.

➔ As noted above, the discrepancy in height is now discussed in Discussion section. Also, we modified the “observed altitude” for the IR2 image and the following sentence was added to this caption:

L.567-569: “Contrast features in the image are believed to reflect the opacity contrast of dense middle and lower clouds with particle size of about $3.7 \mu\text{m}$ (called “mode 3”) located around 49–57 km height.”

Figure 3: Two comments already done previously: 1) Use larger fonts for the colorbars; 2) Add a title/symbol to the colorbar to indicate that they correspond to the vertical velocity, meridional velocity and heat transport.

➔ We enlarged the fonts and added titles/symbols to the color bars.

A question regarding Figures 3e and 3f: from the simulations it appears that the meridional velocity direction may change with longitude and latitude (If I understand well, red color corresponds to meridional flow directed

toward the pole, and blue flow directed toward the equator?). Has this trend (different directions of the meridional flow at the same altitude level) ever been observed?

- ➔ Since these panels show the southern hemisphere, red color corresponds to equatorward flow and blue to poleward flow. There are a few observation (cloud tracked wind) of meridional flows with different directions on the same level (Hueso et al., 2015), but a strong, sharp convergence zone (like as obtained in our simulation) was not found.

Hueso, R., Peralta, J., Garate-Lopez, I., Bandos, T. V., & Sánchez-Lavega, A. (2015). Six years of Venus winds at the upper cloud level from UV, visible and near infrared observations from VIRTIS on Venus Express. *Planetary and Space Science*, 113-114, 78–99. <http://doi.org/10.1016/j.pss.2014.12.010>

Figure 4: Yellow lines: Could you please recheck them. If I understand well, like they are plotted now, they should correspond respectively to a period of about 3.2 Earth days (Fig a) and 7 Earth days (Fig. b). But in the main text it's written they should be 4.6 and 8.2 Earth days. Please add the symbol to the colorbar.

- ➔ Thank you very much for pointing out this inconsistency. The yellow lines in Fig. 4 represent the rotation speed and period at $\text{lat} = -35$, but in the text, those at the equator had been written. We reconsidered and rewrote the text to eliminate the values of the rotation speed and period at the equator, which was not useful for readers.

Figure 5: Caption: I am still not sure to understand what the composite means is. Also, specify that the arrows on the plot correspond to the vectors (u', v') . Add a symbol to the colorbar, larger fonts are needed.

- ➔ Composite mean is described above and also an explanation is added in the text.
We added a symbol to the color bar and enlarged the fonts, and added (u', v') in the caption.

General comment: In order to make the review easier, please add line numbers.

- ➔ We sincerely apologize for lacking the line number. We added the line number in the revised manuscript.

Reviewer #1 (Remarks to the Author):

SUMMARY:

Authors claim that they have reproduced a white long fine (streak) structure observed by Akatsuki's IR2 camera by a simplified version of AFES general circulation model. They relate this streak-structure with downward flow generated by the horizontal convergence of the meridional wind that is reinforced by baroclinic eddies generated in the low-stability layer. These baroclinic disturbance transports heat and angular momentum polewards maintaining the meridional profile of the angular velocity, which is responsible for the tilting of the pressure anomaly associated to an equatorial Rossby-like wave and, therefore, for the extended shape of the region where meridional wind converges.

They check the role of the solar heating and low static stability layer in the formation of the streak-structure by running different experiments. Authors conclude that solar heating is not important for the formation or appearance of the streak-structure. On the contrary, authors find that the low static stability layer plays an important role in the formation of the structure.

In the current version of the manuscript, authors do mention the lifetime of the structure and how the streak structure could be modified in polar regions. They also discussed in detail the possible effects of the altitude discrepancy that I found in the previous version. A new figure showing the static stability profiles for the different runs is added, the use of "streak structure" and "streaks" is improved, and a more detailed discussion about the baroclinic instability is present.

Therefore, I do recommend the publication of the current manuscript after some minor corrections. I thank authors for having followed my suggestions and responded to my questions and comments one by one.

MINOR COMMENTS:

- I think that the use of two different static stability profiles is still a bit confusing. I will suggest the authors to explain the difference between the profile used for the Newtonian cooling and the profile

obtained with the simulated temperature field, and also what profile you refer to when comparing the simulated streak feature and its altitudes.

- Lines 31-33: I think the last sentence would be better understood like this:

The streaks of downward flow result from THE INTERACTION OF THE meridional tilting of the Rossby-like wave's phase lines AND the characteristics of the baroclinic instability produced around the low-stability layer.

- Lines 97-99: I think this sentence would be better understood like this:

Considering THAT the experimental results SHOW the intense baroclinic disturbance only in the cases where the low stability layer exists, ...

- Line 134: The Venus polar vortex shows different shapes; monopole, dipole, irregular, ... Do you see any other shape than dipolar for your simulated vertical velocity's distribution?

- Lines 137-140: At what latitude is the rotation of the streak structure 6 Earth days? Why do you compare the polar vortex's rotation period (latitudes higher than 75deg) with the mean zonal flow's rotation at the equator?

- Lines 147-148: Does blue region mean poleward or equatorward? Do you mention if the meridional wind is defined from south to north or from north to south?

- Lines 169-172: In my opinion these sentences can be confusing. See first comment above.

- Lines 198-199: "Pair of layers" and "pair of isentropic surfaces" refer to the same thing? This is confusing too, can you rewrite it, please?

- Line 228: "We can highlight..." or "The composite means highlights...", right?

- Lines 280-281: "high-latitude zonal jet", observed or simulated?

- Line 281: "The meridional WIND shear ...", right?

- Line 348: I think this sentence would be better understood like this:

"The cloud opacity, FOR EXAMPLE, should give a feedback..."

USE OF ENGLISH:

- Line 27: "Here we succeed, FOR THE FIRST TIME, in reproducing..."

- Lines 44, 47, 50, 59 : AT these wavelengths / AT various infrared wavelengths / AT 2.26microns

- Line 69: "... compared to THAT OF previous studies but CONSIDERING an idealised superrotation..."

- Line 90: "... even without diurnal solar variation."

- Line 117: "... to be reasonably comparable / to be easily compared ..."

- Line 157: "... cooling profile AS in STD ..."

- Line 159: "... (see Supplementary Movie 3), AND polar disturbances ..."

- Line 162: "... streak structure or ITS inter-..." or "... streak structureS or THEIR inter-..."

- Line 162: "This MIGHT be surprising..."

- Line 192: "... ; SLIGHTLY NEGATIVE or even positive ..." ?

- Line 214: "... explained below is largEST in this case." ?

- Line 237: "... degree south latitude WHERE there are ..."

- Line 303: "... seems to BE COMPOSED BY both the ..."

- Line 309: "The situations UNDER WHICH those disturbances TOOK PLACE in the simulated ..."

- Line 317: "HOWEVER, WE keep theSE further investigations..."

- Line 322: "Although there is a REMARKABLE similarity... "

- Line 330: "... flow COMPOSING the streak structure ..."

- Line 342: AT 2.26microns

- Line 378: "... is covered with white both in ..." > "... is white-shaded in both Figs ..." ?

- Line 566 : ON 25 March 2016
 - Line 594: Zonally and temporaLLY
 - Line 608: ... the equatorial valueS of which are ...
 - Line 610: solid line with shading > shaded solid line?
-
- Figure 8: vertical axis: Eight > Height

Reviewer #2 (Remarks to the Author):

This revision represents a big advancement over the original manuscript. The presentation is much clearer, and the authors have addressed all of the major concerns. I do, however, have one remaining problem with current version, as summarized below and described in the attached, annotated PDF.

The problem has to do with the attribution of baroclinic (vs. barotropic) instabilities to the polar phenomena. I should start by saying I have no problem with the presentation of the equatorial waves, and that they provide the equatorward flank of a new style of high-latitude instability that ultimately generates the streak structures in their model. The problem comes with the description of the "Formation mechanism of the streak structure"; the reasoning appears to be somewhat circular and perhaps flawed. The text makes it sound as if the eastward tilting of the eddies are both caused by and responsible for the meridional shear. If this is the authors' intent, then I think there are problems of circularity here. If not, then some significant clarification to the text would be needed. Alternatively, this section could be shortened or removed entirely. The analysis of heat fluxes demonstrates some role for baroclinic instability, but the details of how the streak emerges from them can be left to future work.

I've also included a number of corrections and suggestions for improvement in the attached PDF annotations.

Authors' response to the reviewer#1's comments

Title: Planetary-scale streak structure reproduced in a Venus atmospheric simulation.

Authors: Hiroki Kashimura*, Norihiko Sugimoto, Masahiro Takagi, Yoshihisa Matsuda, Wataru Ohfuchi, Takeshi Enomoto, Kensuke Nakajima, Masaki Ishiwatari, Takao M. Sato, George L. Hashimoto, Takehiko Satoh, Yoshiyuki O. Takahashi, Yoshi-Yuki Hayashi

We thank the reviewer for a careful review and constructive comments, again. Please find below the authors' response. In this reply we denote referee's comments and questions using blue; our responses are in black and relevant text in the manuscript in brown.

First of all, we note that, by following the reviewer#2's major comments, we have removed the last paragraph of the Results section to avoid logical circularity in explanation of the formation mechanism of the streak structure. In the revised manuscript, the last subsection title was changed to "Key features for the formation of the steak structure" and we focused on the differences between ZS0 and ZS4 in characteristic features related to the streak structure in this subsection. Accordingly, the first paragraph of the Discussion section was also revised.

SUMMARY:

Authors claim that they have reproduced a white long fine (streak) structure observed by Akatsuki's IR2 camera by a simplified version of AFES general circulation model. They relate this streak-structure with downward flow generated by the horizontal convergence of the meridional wind that is reinforced by baroclinic eddies generated in the low-stability layer. These baroclinic disturbance transports heat and angular momentum polewards maintaining the meridional profile of the angular velocity, which is responsible for the tilting of the pressure anomaly associated to an equatorial Rossby-like wave and, therefore, for the extended shape of the region where meridional wind converges.

They check the role of the solar heating and low static stability layer in the formation of the streak-structure by running different experiments. Authors conclude that solar heating is not important for the formation or appearance of the streak-structure. On the contrary, authors find that the low static stability layer plays an important role in the formation of the structure.

In the current version of the manuscript, authors do mention the lifetime of the structure and how the streak structure could be modified in polar regions. They also discussed in detail the possible effects of the altitude discrepancy that I found in the previous version. A new figure showing the static stability profiles for the different runs is added, the use of “streak structure” and “streaks” is improved, and a more detailed discussion about the baroclinic instability is present.

Therefore, I do recommend the publication of the current manuscript after some minor corrections. I thank authors for having followed my suggestions and responded to my questions and comments one by one.

MINOR COMMENTS:

- I think that the use of two different static stability profiles is still a bit confusing. I will suggest the authors to explain the difference between the profile used for the Newtonian cooling and the profile obtained with the simulated temperature field, and also what profile you refer to when comparing the simulated streak feature and its altitudes.

➡ We modified and added sentences in the second paragraph of “Importance of the low-stability layer” as follows to explain the difference:

L.170–176: “The static stability to form the low-stability layer given in the basic temperature profile for the Newtonian cooling is increased from 0.1 K km^{-1} of STD/ZS0 to 2.0 K km^{-1} in ZS2 and to 4.0 K km^{-1} in ZS4, respectively as shown by dashed lines in Fig. 3. Note that these given static-stability profiles are used for the basic temperature for the Newtonian cooling, whereas the simulated ones shown by solid lines in Fig. 3 are results of thermal balance among the Newtonian cooling, solar heating, and adiabatic heating/cooling due to atmospheric circulations. In ZS2 and ZS4 the simulated static stability do not reach as low as 1.0 K km^{-1} around 50–60 km heights.”

- Lines 31-33: I think the last sentence would be better understood like this:

The streaks of downward flow result from THE INTERACTION OF THE meridional tilting of the Rossby-like wave's phase lines AND the characteristics of the baroclinic instability produced around the low-stability layer.

➔ The sentence was modified as suggested.

- Lines 97-99: I think this sentence would be better understood like this:

Considering THAT the experimental results SHOW the intense baroclinic disturbance only in the cases where the low stability layer exists, ...

➔ By following the other reviewer's suggestion, we removed the description of the maintenance mechanism. And, sentences around this part were modified as follows:

L.94–97: “The tilting of the equatorial Rossby-like wave seems to be induced by the intense baroclinic disturbances that occur only in the cases where the low stability layer exists. Therefore, we speculate that the streak structure might be a manifestation of front genesis caused by a type of baroclinic instability unique to the Venusian high-latitudes.”

- Line 134: The Venus polar vortex shows different shapes; monopole, dipole, irregular, ... Do you see any other shape than dipolar for your simulated vertical velocity's distribution?

➔ In our simulation, the vertical velocity distribution in the polar region was mostly dipole. It sometimes distorts but it soon recovers to the dipolar shape. Please see the Supplementary Movies.

- Lines 137-140: At what latitude is the rotation of the streak structure 6 Earth days? Why do you compare the polar vortex's rotation period (latitudes higher than 75deg) with the mean zonal flow's rotation at the equator?

➔ The streak structure lying about from latitudes of 30 to 75 degs is rotating with almost the same period of 6 Earth days. Here, we intended to compare the rotation period of the simulated streak structure (~6 Earth days) with that of polar vortex (~ 2–4 Earth days) to say that the polar vortex is different from the streak structure; and then we intended to compare the rotation period of the streak structure with that of the mean zonal flow to say that the streak structure is not just advected by the mean flow. The rotation period of the mean zonal flow is the longest at the equator (~ 4.9 Earth days) and decreases with latitude (i.e., faster in higher latitudes). We recognized that we should have to describe this situation. We divided this sentence into two as follow:

L.137–139: “The rotation period of the simulated dipoles is around 2–4 Earth days, while that of the simulated streak structure is about 6 Earth days. The rotation period of the mean zonal flow at this

level is about 4.9 Earth days at the equator and is shorter in higher latitudes, so that the rotation period of the simulated streak structure is longer than that of the mean flow.”

- Lines 147-148: Does blue region mean poleward or equatorward? Do you mention if the meridional wind is defined from south to north or from north to south?

- ➔ The blue region means poleward. We have not mentioned the direction of positive for the meridional velocity. We added the definition and poleward/equatorward after blue/red as follows:

L.145–148: “The polar view of the meridional velocity (v ; **northward is positive**) field is shown in Fig. 4e, which is at the same level and time as in Fig. 2c. Regions of downward flow stronger than -0.05 m s^{-1} are overlaid by green hatches. A sharp convergence zone of the meridional flow, shown as the boundary between equatorial-sided blue (**poleward**) regions and polar-sided red (**equatorward**) region,”

- Lines 169-172: In my opinion these sentences can be confusing. See first comment above.

- ➔ Please see the reply to the first comment.

- Lines 198-199: “Pair of layers” and “pair of isentropic surfaces” refer to the same thing? This is confusing too, can you rewrite it, please?

- ➔ For here, the other reviewer suggested to use a more standard description and we rewrote as follows:

L.202–206: “As for the emergence of the baroclinic instability, a mean PV-gradient along isentropes must reverse in the vertical direction.³⁵ The isentropic PV-gradient is mostly positive (PV increases with latitude). However, in the mid-latitudes of STD and ZS0, there are regions of negative isentropic PV-gradient on the equator- and lower-side of the PV maxima, which indicates a possibility of baroclinic instability caused by the coupling of disturbances on different levels of isentropes with opposite sign of PV-gradient.”

- Line 228: “We can highlight...” or “The composite means highlights...”, right?

- ➔ Right. We modified the sentence as suggested.

- Lines 280-281: “high-latitude zonal jet”, observed or simulated?

- ➔ Simulated. The word “simulated” was added here.

- Line 281: “The meridional WIND shear ...”, right?
 - ➔ We intended to say “the meridional shear in angular velocity”. This paragraph was removed by following the other reviewer’s suggestion, and in the new sentences, we wrote as follows:
L.289–290: “In terms of angular velocity, a large meridional shear emerges in high-latitudes in ZS0 (Fig. 7c), but not in ZS4 (Fig. 7g).”
- Line 348: I think this sentence would be better understood like this:
“The cloud opacity, FOR EXAMPLE, should give a feedback...”
 - ➔ The sentence was modified as suggested.

USE OF ENGLISH:

- Line 27: “Here we succeed, FOR THE FIRST TIME, in reproducing...”
- Lines 44, 47, 50, 59 : AT these wavelengths / AT various infrared wavelengths / AT 2.26microns
- Line 69: “... compared to THAT OF previous studies but CONSIDERING an idealised superrotation...”
 - ➔ We followed above three corrections.
- Line 90: “... even without diurnal solar variation.”
 - ➔ Here, we changed the wordings to “when simulated without the diurnal cycle of insolation” following the other reviewer’s suggestion.
- Line 117: “... to be reasonably comparable / to be easily compared ...”
- Line 157: “... cooling profile AS in STD ...”
- Line 159: “... (see Supplementary Movie 3), AND polar disturbances ...”
- Line 162: “... streak structure or ITS inter-...” or “... streak structureS or THEIR inter-...”
- Line 162: “This MIGHT be surprising...”
- Line 192: “ ... ; SLIGHTLY NEGATIVE or even positve ...” ?
 - ➔ We followed the all corrections above.
- Line 214: “... explained below is largEST in this case.” ?
 - ➔ We cannot say “largest” here (we did not check it is the largest or not), though it is “enough large”.
- Line 237: “... degree south latitude WHERE there are ...”
- Line 303: “... seems to BE COMPOSED BY both the ...”
- Line 309: “The situations UNDER WHICH those disturbances TOOK PLACE in the simulated ...”

- Line 317: “HOWEVER, WE keep theSE further investigations...”
- Line 322: “Although there is a REMARKABLE similarity... “
- Line 330: “... flow COMPOSING the streak structure ...”
- Line 342: AT 2.26microns
- Line 378: “... is covered with white both in ...” > “... is white-shaded in both Figs ...” ?
 - ➔ We followed the all corrections above.
- Line 566 : ON 25 March 2016
- Line 594: Zonally and temporaLLY
- Line 608: ... the equatorial valueS of which are ...
- Line 610: solid line with shading > shaded solid line?
 - ➔ We followed the all corrections above.
- Figure 8: vertical axis: Eight > Height
 - ➔ It may be a problem of viewer softwares, “height” is shown in our PC. We will carefully check in the proof read.

Authors' response to the reviewer#2's comments

Title: Planetary-scale streak structure reproduced in a Venus atmospheric simulation.

Authors: Hiroki Kashimura*, Norihiko Sugimoto, Masahiro Takagi, Yoshihisa Matsuda, Wataru Ohfuchi, Takeshi Enomoto, Kensuke Nakajima, Masaki Ishiwatari, Takao M. Sato, George L. Hashimoto, Takehiko Satoh, Yoshiyuki O. Takahashi, Yoshi-Yuki Hayashi

We thank the reviewer for a careful review and constructive comments. Please find below the authors' response. In this reply we denote referee's comments and questions using blue; our responses are in black and relevant text in the manuscript in brown.

First of all, we note that we have removed the last paragraph of the Results section to avoid logical circularity in explanation of the formation mechanism of the streak structure by following the reviewer's major comments. In the revised manuscript, the last subsection title was changed to "Key features for the formation of the streak structure" and we focused on the differences between ZS0 and ZS4 in characteristic features related to the streak structure in this subsection. Accordingly, the first paragraph of the Discussion section was also revised.

This revision represents a big advancement over the original manuscript. The presentation is much clearer, and the authors have addressed all of the major concerns. I do, however, have one remaining problem with current version, as summarized below and described in the attached, annotated PDF.

The problem has to do with the attribution of baroclinic (vs. barotropic) instabilities to the polar phenomena. I should start by saying I have no problem with the presentation of the equatorial waves, and that they provide the equatorward flank of a new style of high-latitude instability that ultimately generates the streak structures in their model. The problem comes with the description of the "Formation mechanism of the streak structure"; the reasoning appears to be somewhat circular and perhaps flawed. The text makes it sound as if the eastward tilting of the eddies are both caused by and responsible for the meridional shear. If this is the authors' intent, then I think there are problems of circularity here. If not, then some significant clarification to

the text would be needed. Alternatively, this section could be shortened or removed entirely. The analysis of heat fluxes demonstrates some role for baroclinic instability, but the details of how the streak emerges from them can be left to future work.

- ➔ Thank you very much for this comment. We have carefully considered this and agreed with the reviewer's comment that there was a problem of circularity. We decided to remove the second paragraph in this section. The subsection title was changed to "Key features for the formation of the streak structure" and focused on describing features that differ between ZS0 and ZS4 and seem to be related to the formation of the streak structure. As suggested, details of the baroclinic disturbances and how the streak emerges from them are left to our future work. Details of the revision are described below.

I've also included a number of corrections and suggestions for improvement in the attached PDF annotations.

Reviewer's comments on the PDF:

L31: technics => techniques

L74: of => of the

L90: even when without diurnal solar variation. => , even when simulated without the diurnal cycle of insolation.

- ➔ All above suggestions were followed.

L99: "baroclinic instability" — Perhaps rather than putting "baroclinic instability" in quotes, you could just say "a type of baroclinic instability unique to the Venusian high-latitudes."

- ➔ We rewrote as "one or two types of baroclinic instability unique to the Venusian atmosphere", because there might be two types of baroclinic instability in mid-latitudes and in high-latitudes.

L112: imply that of downward flow. => indicate the presence of strong downward flow.

L173: to => and cannot

L175: further high- => highest

- ➔ All above suggestions were followed.

L179: The — A transition sentence would be helpful to the reader here. For instance: "The true nature of the streak (that is, its formation mechanism) remains to be identified."

➔ We added the following sentence:

L.182: “The true nature of the streak structure (i.e., its formation mechanism) remains to be identified.”

L182: the — remove

➔ We removed “the” as suggested.

L198-199: instability, a pair of layers with opposite sign in the mean meridional PV-gradient is necessary — This wording is a bit confusing. A more standard description is that the PV gradient along isentropes must reverse in the vertical direction, and perhaps you could change this language to conform to that standard.

➔ The sentence was modified as follows:

L.202–203: “As for the emergence of the baroclinic instability, a mean PV-gradient along isentropes must reverse in the vertical direction.”

L199: For in — “In the...”?

L199-200: pair of isentropic surfaces of $\theta = 830$ and 860 K and that of $\theta = 840$ and 870 K could correspond to the opposite PV-gradient layers to => the isentropic PV gradient reverses on either side of the maximum PV, along pairs of isentropes...

L201: another => another region of

L202: could emerge => is present

L202: those => the

L203: positive => positive in ZS2 and ZS4

L204: the — remove

L204: Those => These

L205: the — remove

L206: the structure seems => the resulting structures seem

➔ On the basis of above suggestions, we rewrote the remaining sentences in this paragraph as follows to fit the use of standard description for the necessary condition of the baroclinic instability:

L.203–214: “The isentropic PV-gradient is mostly positive (PV increases with latitude). However, in the mid-latitudes of STD and ZS0, there are regions of negative isentropic PV-gradient on the equator- and lower-side of the PV maxima, which indicates a possibility of baroclinic instability caused by the coupling of disturbances on different levels of isentropes with opposite sign of PV-gradient. Similarly, in the polar regions of STD and ZS0, there appears negative PV-gradient along the isentropes close to the polar PV maxima. In contrast to the low-stability cases, there appear

almost no region of negative isentropic PV-gradient in ZS2 and ZS4; especially in the mid-latitudes, PV increases quite monotonically with latitude. This indicates that the necessary condition for baroclinic instability is not satisfied in those higher-stability cases. These PV trends confirm that the existence of the low-stability layer may cause baroclinic instability as previously reported in simulation studies of Venus atmosphere^{20,21,36}, and suggest that intense baroclinic disturbances are essential for the formation of the streak structure. However, the resulting disturbance structures seem to be more complicated than formerly expected and are left to be investigated in our future work (see Discussion).”

L214: in this case => in amplitude

- ➔ Changing to amplitude will cause repetition of “amplitude”. Here we changed “this case” to “ZS0” to avoid confusion.

L220: of the propagation of — remove

L222: propagates nearly in => propagates with nearly

- ➔ Above 2 suggestions were followed.

L236: Hovmoller — This is the first use of the term Hovmoller, despite having introduced it in the previous paragraph. I suggest defining “Hovmoller plot” when first mentioning Fig. 6a

- ➔ We modified the first sentence of this section as follows:

L.217–218: “A longitude-time cross-section (**as known as Hovmöller diagram**) of pressure anomaly from the zonal mean (p') at 35 degree south latitude and 65 km height in ZS0 is shown in Fig. 6a.”

L240: Remember => Recall

L249: included => included (not shown)

L253: There => In Fig. 7, there

L258: from the date with the => over a

L259: locating roughly along the => roughly along an

L261: the — remove

L263: interpret => suggest ?

L268: wave => wave (Fig. 7)

- ➔ Above 8 suggestions were followed.

L278-281: ZS0. That is, the phase line could be tilted by a large meridional shear of the angular velocity (Fig. 7b), which becomes evident in the polar side edges of the equatorial Rossby-like wave and corresponds to a high-latitude zonal jet (local maximum in the zonal mean zonal wind).

— If I understand this correctly, I don't think I agree with this description. It seems like you're saying high-latitude meridional shear is creating the phase tilting of the (otherwise global) equatorial waves.

If these structures represent instabilities, the high-latitude portion should be thought of as an independent wave-mode that interacts with the equatorial wave. As in your elegant description of the vertical coupling of Kelvin and Rossby waves at the equator, a similar coupling would occur across the meridional/vertical shear at high latitudes for barotropic/baroclinic instability (accounting for the extreme sloping of isentropes).

L281-282: The meridional shear would result from a large amount of poleward angular-momentum transport by eddies shown

— It's confusing to refer to meridional shear being maintained by eddy transports. Because the model is in a steady-state, the divergence of eddy transports must be balanced by some other torque, for instance the Hadley cell.

On Earth, we think of the Hadley cell transporting angular momentum poleward to form the subtropical jet, and eddies carrying the momentum further poleward to an eddy-driven jet. The former mechanism produces a baroclinic jet structure and the latter a barotropic jet structure.

I'm not sure if it's possible to distinguish between these jet structures in your model, because I doubt the lower atmosphere is in steady-state.

Given this (perhaps unavoidable) uncertainty, I would strongly suggest removing this paragraph.

- ➔ As described in the response to the main comment, we decided to remove this paragraph. The title of this section was changed to "Key features for the formation of the streak structure" and we focused on describing the difference between ZS0 and ZS4 in features that may be related to the formation of the streak structure. We believe that describing the features shown in Figs. 7b, c, f, and g is informative for the reader, so that we remained these panels and describe them as follows in this section. Due to the revision, we swapped Fig. 7b(f) and 7c(g) to follow the appearance order in the text.

L.284–293: "The difference in tilting of the eddy circulations between ZS0 and ZS4 is also apparent in angular-momentum transport by eddies shown in Figs. 7b and 7f. A large amount of poleward transport of angular momentum exists in ZS0, whereas the angular momentum transport is very small in ZS4. Correspondingly, another characteristic feature observed in ZS0 contrasted with ZS4 is

the existence of high-latitude zonal jets (local maximum in the zonal mean zonal wind) around 75 degs. as shown by dashed lines in Fig. 7c. In terms of angular velocity, a large meridional shear emerges in high-latitudes in ZS0 (Fig. 7c), but not in ZS4 (Fig. 7g). Recall that the intensity of baroclinic disturbances is strong in ZS0 and quite small in ZS4 as shown in Figs. 5b and 5d, the baroclinic instability must play an important role for producing those features. The structure of the baroclinic disturbances in the model, however, is yet to be fully investigated. We just try to discuss the possible structure briefly in the next section.”

In order to make Discussion consistent with the change in this section, we revised the first paragraph of Discussion as follows:

L.296–318: “Because baroclinic disturbances are associated with strong meridional heat fluxes in the vicinity of zonal jets, we turn to an analysis of these features in our simulations. Figure 8 shows the mean meridional distribution of the north-south eddy heat flux and the mean zonal wind in ZS0. There are two axes of large poleward heat fluxes, each of which implies the emergence of baroclinic instability: one in the mid-latitudes indicated by the white dashed line and the other in the high-latitudes indicated by the black one, in each hemisphere. The locations of the high-latitude heat fluxes align with those of high-latitude jets. This implies that baroclinic eddies in high-latitudes contribute to the formation of the jet by PV stirring^{40,41}, the same manner as that of the mid-latitude jets in the Earth’s troposphere. Then, the high-latitude jets may incline the phase line of the equatorial Rossby-like wave. In contrast to the high-latitudes, contribution of the mid-latitude baroclinic disturbances is unobvious. The axes of jet maxima do not correspond to the maxima of the mid-latitude heat fluxes. Instead, a large amount of eddy angular-momentum fluxes exists over this region as shown in Fig. 7b and indicated by red arrows in Fig. 8 at the altitude of 65 km. We suspect that the lower-level part of the mid-latitude baroclinic disturbances (i.e., Rossby wave) appear as the signals propagating, with a period of about 3 Earth days, faster than the equatorial Kelvin-like wave’s dominant disturbances whose rotation period is about 5.8 Earth day in Fig. 6b. The signals also propagate faster than the mean flow. This is consistent with the direction of the Rossby wave’s phase speed under the negative PV gradient shown by Fig 5f. Then, the following three components might be related to the formation of the streak structure: the high-latitude disturbances and the associated jets, the mid-latitude baroclinic disturbances, and the equatorial Rossby-like waves. The latitudinal angular phase speed of the high-latitude disturbances is the fastest, then that of the mid-latitude disturbance is in the middle, and that of the equatorial disturbance is the slowest. The

differential phase speeds of those three components might cause the meridional tilting of the phase lines of the equatorial Rossby-like wave to form the streak structure; and the north-south symmetry is regulated by the vertical shear instability with the equatorial Rossby-like wave and the equatorial Kelvin-like wave.”

In addition, we modified the first sentence of the second paragraph in Discussion as follows:

L.320–322: “Though we have discussed a possible formation mechanism of the planetary-scale streak structure as above, we should note that the details of the disturbances, instabilities, and the angular momentum balance in our simulation are still unclear and remain to be explored.”

L289-290: In order to consider the role of the baroclinic disturbances, locations of the heat fluxes and jets would be informative => Because baroclinic disturbances are associated with strong meridional heat fluxes in the vicinity of zonal jets, we turn to an analysis of these features in our simulations.

L292: the — remove

➔ Above 2 suggestions were followed.

L294: agree => aligns

➔ Since the subject is plural, we changed to “align” here.

L294: the baroclinic instability => baroclinic eddies

➔ We followed this suggestion.

L330: consisting => consisting of

➔ We modified this to “composing” by following the other reviewer’s suggestion.

L334: at — remove

➔ We followed this suggestion.

L339: fresh air — Please define what you mean by “fresh air”.

➔ Here, we intended to say “the air parcels displaced from the upper clouds”. We changed the sentence to use this phrase instead of “the displaced fresh air parcels”.

L341: larger cloud-particles, so-called “mode 3” => larger, so-called “mode-3” cloud particles

L354: the dynamical process is => large-scale process, rather than radiation or microphysics, are

L365: placed => considered

L368: of => of the

➔ Above 4 suggestions were followed.